# Sample by step, Optimize by Chunk: Chunk-Level GRPO for Text-to-Image Generation

The image is of dancing potatoes in a cute cartoony style.

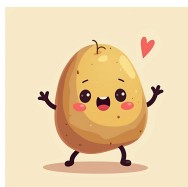 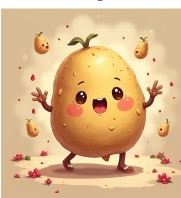 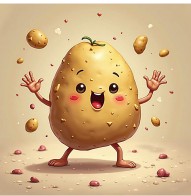 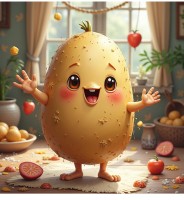 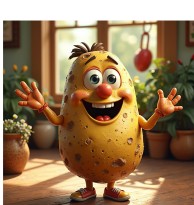

Portrait of an anime princess in white and golden clothes.

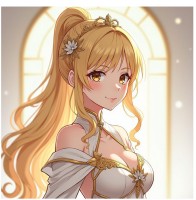 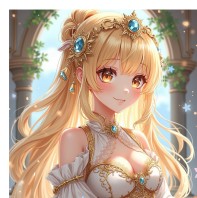 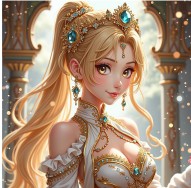 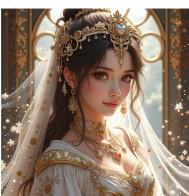 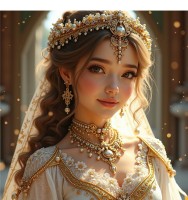

A cute rainbow kitten with different colored eyes in the chibi-style of Studio Ghibli is featured on a postcard.

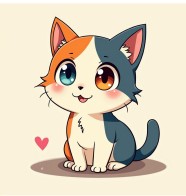 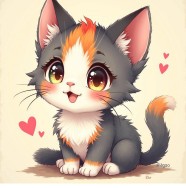 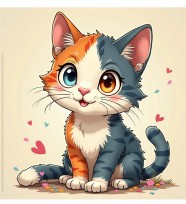 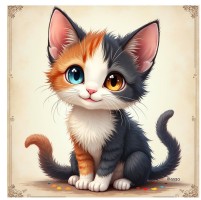 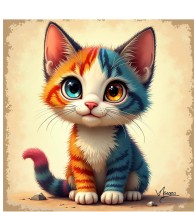

| FLUX | DanceGRPO | Chunk-GRPO w/o temporal dynamics | Chunk-GRPO w/ temporal dynamics | Chunk-GRPO w/ weighted sampling |

Figure 1: Chunk-GRPO significantly improves image quality, particularly in structure, lighting, and fine-grained details, demonstrating the superiority of chunk-level optimization.

## ABSTRACT

Group Relative Policy Optimization (GRPO) has shown strong potential for flow-matching-based text-to-image (T2I) generation, but it faces two key limitations: inaccurate advantage attribution, and the neglect of temporal dynamics of generation. In this work, we argue that shifting the optimization paradigm from the step level to the chunk level can effectively alleviate these issues. Building on this idea, we propose Chunk-GRPO, the first chunk-level GRPO-based approach for T2I generation. The insight is to group consecutive steps into coherent 'chunk's that capture the intrinsic temporal dynamics of flow matching, and to optimize policies at the chunk level. In addition, we introduce an optional weighted sampling strategy to further enhance performance. Extensive experiments show that Chunk-GRPO achieves superior results in both preference alignment and image quality, highlighting the promise of chunk-level optimization for GRPO-based methods.

# 1 INTRODUCTION

Reinforcement learning (RL)(Sutton et al., 1998; Schulman et al., 2017) has recently found success beyond traditional domains, particularly in the reasoning of Large Language Models (LLMs)(Jaech et al., 2024; Guo et al., 2025). Inspired by these advances, recent works(Xue et al., 2025; Liu et al., 2025b; Wang & Yu, 2025) have explored applying RL to text-to-image (T2I) generation for aligning specific preferences. In this context, Group Relative Policy Optimization (GRPO)(Shao et al., 2024; Guo et al., 2025) has emerged as a promising approach for flow-matching-based T2I generation (Lipman et al., 2022; Liu et al., 2023; Esser et al., 2024). GRPO-based methods typically sample a group of images from the same prompt, evaluate them using reward models, convert the rewards into group relative advantages, and assign these advantages equally across all timesteps for optimization.

While effective, this uniform assignment intro-
duces two key limitations: (1) inaccurate ad-
vantage attribution, and (2) disregard for the
temporal dynamics of generation. We first il-
lustrate the former in Figure 2, and discuss tem-
poral dynamics later. Consider two generation
trajectories from the same prompt in Figure 2,
each consisting of three timesteps. Although
the final advantage correctly favors the better
trajectory ($Trajectory_1$), assigning this same
advantage uniformly across all timesteps incor-
rectly assumes that every step in $Trajectory_1$
is superior to its counterpart in $Trajectory_2$.
However, at timestep $t = 1$ $Trajectory_2$
is clearly better than $Trajectory_1$, despite
$Trajectory_1$ achieving the higher overall re-
ward.

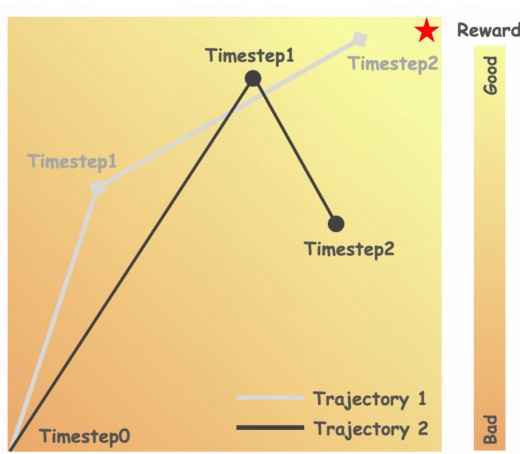

Figure 2: While $Trajectory_1$ has greater final re-
ward (advantage), its $t = 1$ timestep is worse than
that in $Trajectory_2$. However, GRPO assigns the
final advantages equally across all timesteps.

To address this, we draw inspiration from ac-
tion chunking (Zhao et al., 2023; Li et al.,
2025b) in robotics, which predicts sequences
of consecutive actions jointly rather than treat-
ing each step independently. In a similar spirit, we propose to group consecutive timesteps into 'chunk's, and optimize at the chunk level rather than the step level. This alleviates the issue of inaccurate advantage attribution, as we analyze in detail in Section 4.1. Related ideas have been explored in LLMs as Group Sequence Policy Optimization (GSPO) (Zheng et al., 2025), where an entire token sequence is treated as a single unit (analogous to viewing the whole trajectory as one chunk). However, our preliminary studies reveal that different chunk settings (e.g. how many consecutive timesteps for a chunk) have a substantial impact on performance.

We argue that this is due to the overlooking of temporal dynamics of flow matching generation, which we proposed before. Different from LLMs, flow matching exhibits distinct temporal dynam-
ics: each timestep operates under different noise conditions and contributes differently to the final image. Specifically, following (Wimbauer et al., 2024; Liu et al., 2025a) , we analyze the relative $L1$ distance of intermediate latents. As shown in Figure 3, the results reveal clear, prompt-invariant dynamic patterns that naturally segment the trajectory into meaningful chunks. These observations suggest that chunks should not be arbitrary but guided by the inherent temporal dynamics, with timesteps that are dynamically correlated optimized together.

Based on these, we propose Chunk-GRPO, a novel chunk-level RL approach for flow-matching-
based T2I generation. As demonstrated in Figure 4, our key innovation is grouping timesteps into chunks that reflect temporal dynamics, and optimizing them as units with a principled chunk-level importance ratio. Furthermore, motivated by the varying contributions of different chunks, we de-
sign an optional weighted sampling strategy to further boost Chunk-GRPO's performance.

Our contributions can be summarized as follows:

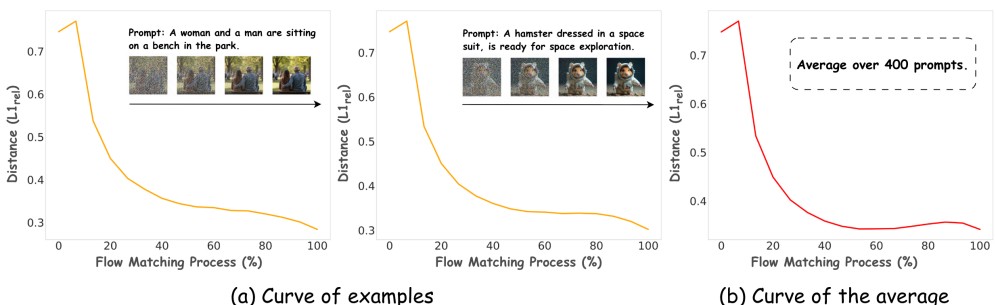

(a) Curve of examples            (b) Curve of the average

Figure 3: The prompt-invariant temporal dynamics of flow matching.

- We are the first to introduce the chunk-level RL optimization for T2I generation. We pinpoint that chunk-level optimization alleviates the inaccurate advantage attribution and mitigates the neglect of temporal dynamics from GRPO-based approaches.

- We propose Chunk-GRPO, a novel chunk-level approach for flow-matching-based T2I generation, which integrates chunk-level optimization with temporal-dynamic-guided chunking. An optional weighted sampling strategy is introduced to push Chunk-GRPO further.

- Extensive experiments demonstrate that Chunk-GRPO achieves superior performance on preference alignment and standard T2I benchmarks, highlighting the effectiveness of chunk-level optimization.

## 2 RELATED WORK

We provide a brief introduction to action chunking in this section, while the remaining related work is discussed in Section B.

### 2.1 ACTION CHUNK

Action chunking (Zhao et al., 2023; Lai et al., 2022) has been widely applied to robotics Chi et al. (2023). This approach mitigates compounding error and non-Markovian noise in human demonstrations by jointly predicting a sequence of future actions rather than a single step. By shortening the effective control horizon, action chunking enables smoother and more stable rollouts. Recently, it has also proven effective in vision-language-action (VLA) models (Black et al., 2024a; Intelligence et al.) and in RL (Li et al., 2025b). These successes suggest that chunking stabilizes long-horizon prediction, accelerates value propagation, and more effectively leverages non-Markovian behavior.

## 3 PRELIMINARY

### 3.1 FLOW MATCHING

Suppose that $x_0 \sim \mathbb{X}_0$ is a data sample from the true distribution, and $x_1 \sim \mathbb{X}_1$ is a noise sample. Following (Liu et al., 2023), the intermediate noised samples $x_t$ can be expressed as:

$$x_t = (1 - t)x_0 + tx_1, \tag{1}$$

where $t \in [0, 1]$ denotes the noise level. Then, flow matching aims to directly regress the estimated velocity field $\hat{v}_\theta(x_t, t)$ by minimizing the objective function (Lipman et al., 2022):

$$\mathcal{L}_{\text{FM}}(\theta) = \mathbb{E}_{t, x_0 \sim \mathbb{X}_0, x_1 \sim \mathbb{X}_1}[\|v - \hat{v}_\theta(x_t, t)\|_2^2], \tag{2}$$

where $v = x_1 - x_0$ represents the target velocity field. Furthermore, a deterministic Ordinary Differential Equation (ODE) is utilized to model the forward process of flow matching:

$$dx_t = \hat{v}_\theta(x_t, t)dt. \tag{3}$$

## 3.2 GRPO ON FLOW MATCHING

As an RL algorithm, GRPO (Guo et al., 2025; Shao et al., 2024) effectively eliminates the need for an additional critic model by estimating the baseline through group-wise relative rewards. In line with the settings of DDPO (Black et al., 2024b), GRPO is also applied in flow matching. Given a group of $G$ images $\{x_0^i\}_{i=1}^G$ generated from the same prompt $c$, the advantage corresponding to the $i$-th sample is formulated as:

$$A_t^i = \frac{r(x_0^i, c) - \text{mean}(\{r(x_0^j, c)\}_{j=1}^G)}{\text{std}(\{r(x_0^j, c)\}_{j=1}^G)}. \tag{4}$$

Notice that $A_t^i$ always keeps the same for any timestep $t$. For simplicity, we neglect the subscript and denote it as $A^i$. The policy is updated by maximizing the following GRPO objective:

$$J(\theta) = E_{c, \{x^i\}_{i=1}^G}$$
$$\left[ \frac{1}{G}\frac{1}{T} \sum_{i=1}^G \sum_{t=1}^T \left( min\left(r_t^i\left(\theta\right) A^i, clip\left(r_t^i\left(\theta\right), 1-\epsilon, 1+\epsilon\right) A^i\right) - \beta D_{KL}\left(\pi_\theta || \pi_{ref}\right) \right) \right], \tag{5}$$

Where $r_t^i$ denotes the importance ratio:

$$r_t^i(\theta) = \frac{p_\theta(x_{t-1}^i | x_t^i, c)}{p_{\text{old}}(x_{t-1}^i | x_t^i, c)}. \tag{6}$$

Furthermore, to meet the exploration requirement of RL, Flow-GRPO (Liu et al., 2025b) and Dance-GRPO (Xue et al., 2025) introduce stochasticity into flow matching by transforming the deterministic ODE into an equivalent Stochastic Differential Equation (SDE):

$$dx_t = \left(v_\theta(x_t, t) + \frac{\sigma_t^2}{2t}(x_t + (1-t)v_\theta(x_t, t))\right)dt + \sigma_t dw_t, \tag{7}$$

where $dw_t$ represents the increments of the Wiener process and $\sigma_t$ controls the stochasticity.

# 4 METHOD

In this section, we begin by introducing chunk-level optimization for GRPO and show why it improves upon standard step-level GRPO in Section 4.1. Next, we describe how to set chunks using the inherent temporal dynamics of flow matching in Section 4.2. Finally, we present our proposed Chunk-GRPO along with an optional weighted sampling strategy in Section 4.3.

## 4.1 CHUNK-LEVEL OPTIMIZATION FOR GRPO

Recall the example in Figure 2. With standard step-level GRPO loss in Equation (5), the optimization object for timesteps $t = 1$ and $t = 2$ in the two trajectories is:

$$J(\theta) = \frac{1}{G}\frac{1}{T} \sum_{i=1}^2 \sum_{t=1}^2 \left( min\left(r_t^i\left(\theta\right) A^i, clip\left(r_t^i\left(\theta\right), 1-\epsilon, 1+\epsilon\right) A^i\right) - \beta D_{KL}\left(\pi_\theta || \pi_{ref}\right) \right). \tag{8}$$

As discussed in Section 1, this uniform stepwise assignment introduces inaccurate advantage attribution. To alleviate this, the first principle of chunk-level optimization is to group consecutive timesteps into chunks and optimize each chunk as a unit. In the example case, the optimization then becomes:

$$J(\theta) = \frac{1}{G} \sum_{i=1}^2 \left( min\left(r^i\left(\theta\right) A^i, clip\left(r^i\left(\theta\right), 1-\epsilon, 1+\epsilon\right) A^i\right) - \beta D_{KL}\left(\pi_\theta || \pi_{ref}\right) \right), \tag{9}$$

where the importance ratio is redefined over the chunk likelihood:

$$r^i(\theta) = \left( \prod_{t=1}^2 \frac{p_\theta\left(x_{t-1}^i | x_t^i, c\right)}{p_{\text{old}}\left(x_{t-1}^i | x_t^i, c\right)} \right)^{\frac{1}{2}}. \tag{10}$$

The key underlying proposition is as follows:

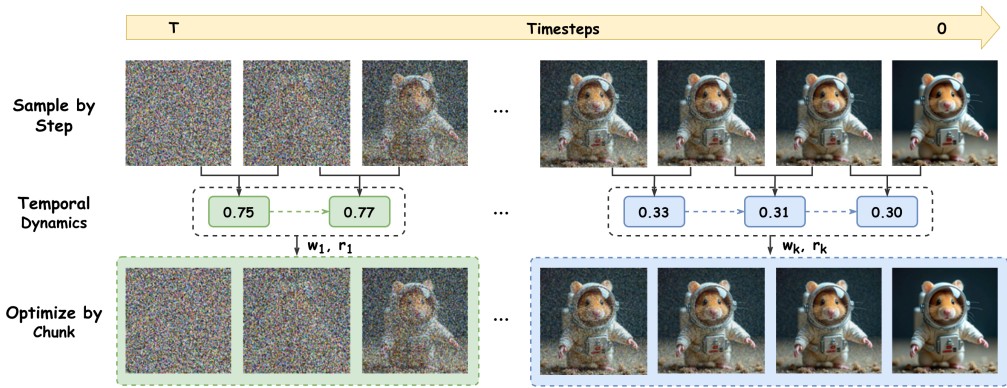

Figure 4: The overall framework of Chunk-GRPO. Chunk-GRPO integrates chunk-level optimization with temporal-dynamic-guided chunking, based on the grounded defined chunk-level importance ratio $r$. It also introduces an optional weighted sampling strategy, assigning sampling weight $w$ for each chunk.

**Proposition 1.** *When advantage attribution is inaccurate at individual timesteps, optimizing them jointly within chunk yields better performance than optimizing them independently as steps, especially when the chunk size is small(e.g. a chunk size of 5).*

A mathematical analysis is provided in Section A. With this insight, we formally define chunk-level optimization for GRPO as follows: Given an image generation trajectory:

$$(x_T, x_{T-1}, \cdots, x_2, x_1, x_0)^i, \tag{11}$$

we split it into $K$ different chunks [1]:

$$\{ch_1, ch_2, \cdots, ch_K\}^i = \{(x_T, \cdots, x_{T-cs_1+1}), (x_{T-cs_1}, \cdots, x_{T-cs_1-cs_2+1}), \cdots, (\cdots, x_1)\}^i,$$

$$\sum_{j=1}^{k} cs_j^i = T, \tag{12}$$

where $cs_j$ denotes the chunk size of the $j$-th chunk $ch_j$. The chunk-level optimization objective is then:

$$J(\theta) = E_{c, \{x^i\}_{i=1}^G}$$

$$\left[ \frac{1}{G} \frac{1}{K} \sum_{i=1}^{G} \sum_{j=1}^{K} \left( min \left( r_j^i(\theta) A^i, clip \left( r_j^i(\theta), 1-\epsilon, 1+\epsilon \right) A^i \right) - \beta D_{KL} \left( \pi_\theta || \pi_{ref} \right) \right) \right], \tag{13}$$

where we redefine the importance ratio $r_j^i(\theta)$ based on chunk likelihood:

$$r_j^i(\theta) = \left( \prod_{t \in ch_j} \frac{p_\theta \left( x_{t-1}^i | x_t^i, c \right)}{p_{old} \left( x_{t-1}^i | x_t^i, c \right)} \right)^{\frac{1}{cs_j}}. \tag{14}$$

Thus, optimization shifts from step-level to chunk-level, alleviating the issue of inaccurate advantage attribution. Notably, setting $K = 1$ will group the whole trajectory into a single chunk, and the optimization further shifts to sequence-level similar to GSPO (Zheng et al., 2025). Conversely, setting $K = T$ will force $cs_j = 1$, and the optimization reverts to standard step-level GRPO.

Using Equation (13) and Equation (14), we have shifted the optimization from step-level to chunk-level. However, the central question then becomes: given the many possible chunk configurations ($ch_j$ and $cs_j$), how should chunks be determined?

---

[1] we neglect $x_{-1}^i$ because there is no more transition into $x_{-1}^i$

## 4.2 CHUNK WITH TEMPORAL DYNAMICS

Before diving into the deeper analysis, we first designed a toy experiment, where all chunks are fixed with an equal chunk size $cs_1 = cs_2 \cdots = cs_k$. As shown in Figure 5, performance varies with chunk size, indicating that chunk design is non-trivial.

We attribute this to the inherent temporal dynamics of flow matching. Unlike LLMs, flow matching consists of time-dependent dynamics in the generation process, where different timesteps contribute unequally to image quality. To better understand this, following (Wimbauer et al., 2024; Liu et al., 2025a), we illustrate the relative $L1$ distance $L1_{\mathrm{rel}}(x, t)$ throughout the generation process:

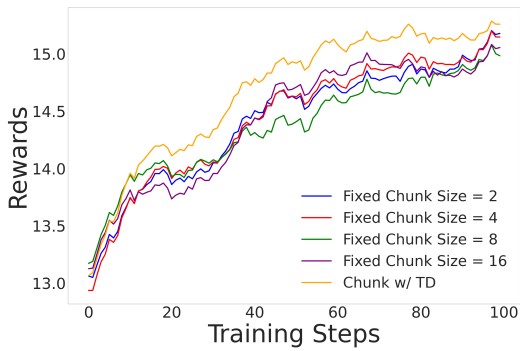

Figure 5: Performance varies with different chunk sizes. The 'TD' refers to temporal dynamics.

$$L1_{\mathrm{rel}}(x, t) = \frac{\|x_t - x_{t-1}\|_1}{\|x_t\|_1}. \tag{15}$$

As shown in Figure 3, $L1_{\mathrm{rel}}(x, t)$ exhibits prompt-invariant yet timestep-dependent patterns. A large $L1_{\mathrm{rel}}(x, t)$ indicates rapid latent changes, while a small value indicates that adjacent latents are similar to each other. From this observation, we argue that: Timesteps with similar dynamics should be grouped into the same chunk, while timesteps with different dynamics should be separated into different chunks.

Fortunately, the prompt-invariant dynamic patterns of $L1_{\mathrm{rel}}(x, t)$ naturally segment the trajectory into meaningful chunks, yielding temporal-dynamic-guided chunks. Thus, we can set chunks based on the relative $L1$ distance, aligning the optimization process with the intrinsic temporal structure of flow matching.

## 4.3 CHUNK-GRPO

We now present Chunk-GRPO, which integrates chunk-level optimization with temporal-dynamic-guided chunking.

Specifically, given an image generation trajectory, we first compute the relative $L1$ distance and set chunks like Equation (12) according to the dynamic profile. This yields the chunk numbers $K$ and chunk sizes $cs_j$. The optimization then follows the chunk-level object in Equation (13). The whole framework is shown in Figure 4.

In practice, we observe that the choice of $K$ and $cs_j$ is closely tied to the total number of sampling steps $T$. A practical strategy, which we adopt in our experiment, is to precompute chunk boundaries based on observed dynamics and keep them fixed throughout training. A detailed discussion is provided in Section 5.1 and Section C.1.

Furthermore, we propose an optional weighted sampling strategy to further enhance Chunk-GRPO. Following Dance-GRPO (Xue et al., 2025), we select only a fraction of chunks (e.g. with fraction 0.5) per update; but instead of uniform sampling, we assign sampling weight $w$ for each chunk:

$$w(ch_j) = \frac{\frac{1}{cs_j} \sum_{t \in ch_j} L1_{\mathrm{rel}}(x, t)}{\frac{1}{T} \sum_{t=1}^{T} L1_{\mathrm{rel}}(x, t)}. \tag{16}$$

From Figure 3, this strategy biases the sampling process toward high-noise regions, and the motivation primarily stems from our ablation studies on training specific chunks. However, although this strategy improves certain aspects of Chunk-GRPO, its overall effects on image quality remain nuanced, as discussed in Section 5.3.

## 5 EXPERIMENTS

### 5.1 EXPERIMENT SETUP

**Training Settings.** We adopt Dance-GRPO (Xue et al., 2025) as the baseline and conduct experiments with FLUX.1 Dev (Labs, 2024) as our base model. HPDv2.1 (Wu et al., 2023) serves as the dataset, while HPSv3 (Ma et al., 2025) is used as the primary reward model. In ablation studies Section 5.3, we additionally validate our approach with Pick Score (Kirstain et al., 2023) and CLIP (Radford et al., 2021) as the reward model. For the chunk setting, we use $\{cs_j\}_{j=1}^4 = [2, 3, 4, 7]$ with total sampling steps $T = 17$ [2], fixed throughout training. Further explanation of chunk configuration, an adaptive chunking strategy, and additional training details are provided in Section C.

**Evaluation Details.** We evaluate both preference alignment and standard T2I benchmarks. For preference alignment, we use HPSv3 (Ma et al., 2025) and ImageReward (Xu et al., 2023) as in-domain and out-of-domain evaluation metrics, respectively, on the HPDv2.1 (Wu et al., 2023) test set. For the standard T2I benchmark, we report results on WISE (Niu et al., 2025), using its rewritten version due to its improved generalization. We also report results on GenEval (Ghosh et al., 2023) in ablation studies Section 5.3. All evaluations adopt hybrid inference from (Li et al., 2025a), which has proven effective in mitigating reward hacking. More details are provided in Section C.3.

### 5.2 MAIN RESULTS

Table 1 presents results on preference alignment, and Table 2 shows WISE benchmark results. Chunk-GRPO consistently outperforms both the base model and Dance-GRPO, confirming the effectiveness of chunk-level optimization. Qualitative comparisons in Figure 1 and Section E further highlight Chunk-GRPO's improvements in image quality. Chunk-GRPO generates outputs that align more closely with human aesthetic

Table 1: Results on Preference Alignment

| Model | HPSv3 | ImageReward |
|---|---|---|
| Flux | 13.804 | 1.086 |
| Dance-GRPO | 15.080 | 1.141 |
| Chunk-GRPO w/o ws | 15.236 | 1.147 |
| Chunk-GRPO w/ ws | 15.373 | 1.149 |

[1] The 'ws' refers to the weighted sampling strategy.

preferences, exhibiting stronger lighting contrast, more vivid colors, and finer details.

For preference alignment, our approach achieves significant additional gains of up to 23% over the baseline across both in-domain and out-of-domain evaluations. On WISE, our approach achieves the strongest overall performance. We attribute the moderate improvement on WISE to a misalignment between the evaluation task and the reward model. Specifically, WISE focuses on reasoning capabilities in T2I generation, whereas HPSv3 is a reward model optimized primarily for aesthetic alignment. To further validate the effectiveness of Chunk-GRPO, we conducted experiments on the standard GenEval benchmark in Section 5.3, where we observed significantly larger improvements.

Notably, the weighted sampling strategy enhances preference alignment but has mixed effects on WISE, a phenomenon we further analyze in Section 5.3.

Table 2: Results on WISE

| Model | Cultural | Time | Space | Biology | Physics | Chemistry | Overall |
|---|---|---|---|---|---|---|---|
| Flux | 0.75 | 0.70 | 0.76 | 0.69 | 0.71 | 0.68 | 0.73 |
| Dance-GRPO | 0.82 | 0.75 | 0.78 | 0.66 | 0.69 | 0.64 | 0.75 |
| Chunk-GRPO w/o ws | 0.82 | 0.76 | 0.77 | 0.68 | 0.69 | 0.68 | 0.76 |
| Chunk-GRPO w/ ws | 0.80 | 0.73 | 0.76 | 0.64 | 0.65 | 0.62 | 0.73 |

[1] The 'ws' refers to the weighted sampling strategy.
[2] We use the rewritten version of WISE.

---

[2] We neglect the last timestep following Dance-GRPO, as the last step does not introduce stochasticity.

Table 3: Ablation Results of Chunk Setting

| Model | Sampling Timesteps | Chunk Setting | HPSv3 |
|---|---|---|---|
| Flux | - | - | 13.804 |
| DanceGRPO | 17 | - | 15.080 |
| | 25 | - | 15.015 |
| Chunk-GRPO w/o td | 17 | $[2, 2, \cdots, 2]$ | 15.115 |
| | 17 | $[4, 4, 4, 4]$ | 15.078 |
| | 17 | $[8, 8]$ | 15.173 |
| | 17 | $[16]$ | 15.142 |
| | 25 | $[2, 2, \cdots, 2]$ | 15.057 |
| | 25 | $[4, 4, \cdots, 4]$ | 15.136 |
| | 25 | $[12, 12]$ | 15.111 |
| | 25 | $[24]$ | 15.100 |
| Chunk-GRPO w/ td | 17 | $[2, 3, 4, 7]$ | 15.236 |

[1] The 'td' refers to the temporal dynamics.

## 5.3 ABLATION STUDY

**Chunk Setting.** We first vary chunk settings under different total sampling steps, excluding the weighted sampling strategy to isolate chunk setting effects. Results in Table 3 show that chunk-level optimization consistently outperforms standard step-level GRPO.

Moreover, temporal-dynamics-guided chunking outperforms fixed chunk size, underscoring the importance of aligning the optimization process with the intrinsic temporal structure of flow matching.

**Training on Specific Chunks.** We next train Chunk-GRPO on individual chunks only. Note that we also remove the weighted sampling strategy here. Results in Figure 6 show that high-noise chunks (e.g., $ch_1$) yield larger improvements than low-noise chunks (e.g., $ch_4$), but they suffer from training instability (e.g., after 60 steps). This motivated our weighted sampling strategy in Equation (16), which adaptively emphasizes high-noise chunks.

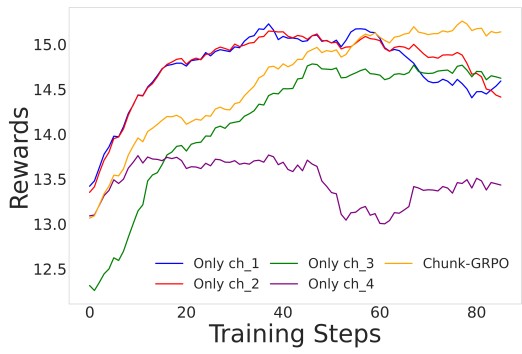

Figure 6: The results of training specific chunks.

**Weighted Sampling Strategy.** As shown in Table 1 and Table 2, the optional weighted sampling strategy improves preference alignment but slightly reduces WISE performance. Careful qualitative analysis reveals a trade-off: while the strategy accelerates preference optimization, it can destabilize image structure in high-noise regions, occasionally leading to semantic collapse. A failure example is shown in Figure 7. Although all methods struggle with this challenging prompt (e.g. Dance-GRPO misses the attribute 'sleeveless'), the weighted sampling strategy further alters the overall image structure, producing the worst case by omitting the entire item 'black loafers' and only partially showing 'capris'). This demonstrates the complex effects of the strategy.

**Reward Models.** Finally, we test Chunk-GRPO's robustness under different reward models. We first replace Hpsv3 with Pick Score (Shukor et al., 2025) as our reward model. Results in Table 4 confirm that Chunk-GRPO consistently outperforms standard step-level GRPO regardless of the reward model, validating its generality.

Since both HPSv3 and PickScore are reward models primarily designed for preference alignment, we further validate our approach using CLIP (Radford et al., 2021), which, while not a preference alignment model, is well recognized for its ability to capture high-level semantics.

A painting of a woman by Zinaida Serebriakova wearing a T-shirt with the Supreme brand logo, a sleeveless white blouse, dark brown capris, and black loafers.

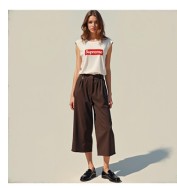 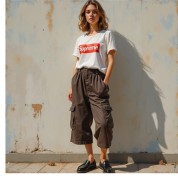 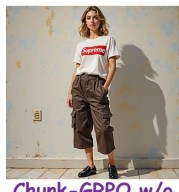 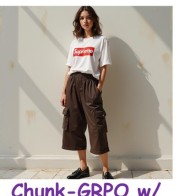 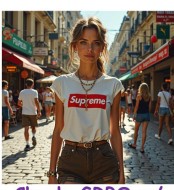

FLUX          DanceGRPO     Chunk-GRPO w/o    Chunk-GRPO w/    Chunk-GRPO w/
                            temporal dynamics  temporal dynamics  weighted sampling

Figure 7: A failure case of the weighted sampling strategy. The strategy wrongly changes the image structure in the high-noise region, leading to the worst variant.

We evaluate it on GenEval(Ghosh et al., 2023), a benchmark that mainly tests instruction-following capabilities of T2I generation. Results in Table 5 demonstrate that Chunk-GRPO also significantly outperforms standard step-level GRPO, demonstrating its broader generalization and robustness beyond preference alignment tasks. Specifically, Chunk-GRPO achieves a performance gain of 0.03, which is 3 times larger than the gain achieved by Dance-GRPO. It is worth noting that the weighted sampling strategy results in a decline in GenEval's semantic performance, which further corroborates our previous analysis.

Table 4: Ablation on Different Reward Models

| Model | Pick Score | HPSv3 | Image Reward |
|---|---|---|---|
| Flux | 22.643 | 13.804 | 1.086 |
| DanceGRPO | 23.427 | 14.612 | 1.208 |
| Chunk-GRPO w/o ws | 23.442 | 14.810 | 1.222 |
| Chunk-GRPO w/ ws | 23.476 | 14.913 | 1.233 |

[1] The 'ws' refers to the weighted sampling strategy.

Table 5: Results on GenEval

| Model | Single Obj. | Two Obj. | Counting | Colors | Position | Color Attr. | Overall |
|---|---|---|---|---|---|---|---|
| Flux | 0.99 | 0.83 | 0.71 | 0.75 | 0.24 | 0.44 | 0.66 |
| Dance-GRPO | 1.00 | 0.86 | 0.71 | 0.78 | 0.22 | 0.46 | 0.67 |
| Chunk-GRPO w/o ws | 0.99 | 0.85 | 0.75 | 0.81 | 0.21 | 0.51 | 0.69 |
| Chunk-GRPO w/ ws | 0.98 | 0.82 | 0.73 | 0.76 | 0.27 | 0.48 | 0.67 |

[1] The 'ws' refers to the weighted sampling strategy.

## 6 CONCLUSION

In this paper, we propose Chunk-GRPO, the first chunk-level GRPO-based approach for flow-matching-based T2I generation. By leveraging the temporal dynamics of flow matching, Chunk-GRPO groups consecutive timesteps into chunks and optimizes at the chunk level, achieving consistent improvements over standard step-level GRPO. We further introduce an optional weighted sampling strategy to push Chunk-GRPO further.

Despite its strong performance, several limitations remain. First, exploring how to combine heterogeneous rewards across different chunks (e.g., employing different reward models for high- vs. low-noise regions) could unlock further improvements. Second, our chunk segmentation remains fixed throughout training. Developing self-adaptive or dynamic chunking strategies that adjust to training signals would be an important next step.

## LLM USAGE

We employed LLMs solely for polishing the language of the manuscript. All scientific contributions, ideas, and analyzes are entirely the authors' own.

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

# A   MATHEMATICAL ANALYSIS

Here we Here we provide a mathematical analysis for Proposition 1. For simplicity, we assume that there are $m$ timesteps with inaccurate advantage attribution between two trajectory segments:

$$(x_T, x_{T-1}, \cdots, x_2, x_1, x_0)^1,$$
$$(x_T, x_{T-1}, \cdots, x_2, x_1, x_0)^2, \tag{17}$$

where $1 \le m \le T$. We denote $T_a$ and $T_{ia}$ as the sets of timesteps with accurate and inaccurate advantage attribution, respectively, and: [3]

$$T_a \cap T_{ia} = \emptyset, \quad T_a \cap T_{ia} = \{1, 2, \cdots, T\}. \tag{18}$$

Let $A^i$ and $A^j$ as the advantage of the two trajectories. Without loss of generality, we assume:

$$A^1 = 1, \quad A^2 = -1. \tag{19}$$

We denote $\hat{A}_t^i$ as the ground-truth advantage. Then for each timestep $t$:

$$\hat{A}_t^1 = A_t^1 = 1, \quad \hat{A}_t^2 = A_t^2 = -1, \quad t \in T_a,$$
$$\hat{A}_t^1 = -A_t^1 = -1, \quad \hat{A}_t^2 = -A_t^2 = 1, \quad t \in T_{ia}. \tag{20}$$

The expected ground-truth loss object can thus be expressed as:

$$J(\hat{\theta}) = \sum_{i=1}^{2} \sum_{t=1}^{T} min\left(r_t^i(\theta)\hat{A}^i, clip\left(r_t^i(\theta), 1 - \epsilon, 1 + \epsilon\right)\hat{A}^i\right). \tag{21}$$

Here we omit constant factors such as $\frac{1}{2}$, $\frac{1}{T}$, and KL KL regularization. The step-level importance ratio $r_t^i$ is defined in Equation (6), reproduced here for clarity::

$$r_t^i(\theta) = \frac{p_\theta(x_{t-1}^i | x_t^i, c)}{p_{old}(x_{t-1}^i | x_t^i, c)}. \tag{22}$$

Substituting Equation (20) into eq. (21), we obtain:

$$J(\hat{\theta}) = \sum_{t \in T_a} \left[min\left(r_t^1(\theta), clip\left(r_t^1(\theta), 1 - \epsilon, 1 + \epsilon\right)\right) + min\left(-r_t^2(\theta), -clip\left(r_t^2(\theta), 1 - \epsilon, 1 + \epsilon\right)\right)\right]$$
$$+ \sum_{t \in T_{ia}} \left[min\left(r_t^2(\theta), clip\left(r_t^2(\theta), 1 - \epsilon, 1 + \epsilon\right)\right) + min\left(-r_t^1(\theta), -clip\left(r_t^1(\theta), 1 - \epsilon, 1 + \epsilon\right)\right)\right]. \tag{23}$$

Since the clipping operation only affects timesteps where the importance ratio lies outside the trust region (Schulman et al., 2015), and such cases are rare under small policy updates, we approximate the gradient of Equation (23) by the gradient of following expression:

$$J(\hat{\theta}) = \sum_{t \in T_a} \left(r_t^1(\theta) - r_t^2(\theta)\right) + \sum_{t \in T_{ia}} \left(r_t^2(\theta) - r_t^1(\theta)\right). \tag{24}$$

Similarly, the step-level GRPO loss has gradient approximated to the gradient of following:

---

[3]we neglect $x_0$ because there is no more transition into $x_{-1}$

$$J(\theta)_{GRPO} = \sum_{t \in T_a} \left( r_t^1(\theta) - r_t^2(\theta) \right) + \sum_{t \in T_{ia}} \left( r_t^1(\theta) - r_t^2(\theta) \right). \tag{25}$$

We now analyze chunk-level optimization. For simplicity, we treat each trajectory in Equation (17) as a single chunk. Following Equation (12), we have:

$$\{ch_1\}^i = \{(x_T, \cdots, x_1)\}^i, \quad i = 1, 2, \tag{26}$$
$$cs_1^1 = cs_1^2 = T,$$

The reason is that if trajectories are split into smaller chunks, each chunk can be viewed as a complete trajectory as in Equation (17). For convenience, we rewrite the chunk-level importance ratio from Equation (14) as:

$$s_j^i(\theta) = \left( \prod_{t \in ch_j} \frac{p_\theta\left(x_{t-1}^i | x_t^i, c\right)}{p_{\theta_{old}}\left(x_{t-1}^i | x_t^i, c\right)} \right)^{\frac{1}{cs_j}}. \tag{27}$$

The chunk-level objective then becomes:

$$J(\theta)_{chunk} = \sum_{i=1}^{2} min\left( s_1^i(\theta) A^i, clip\left( s_1^i(\theta), 1 - \epsilon, 1 + \epsilon \right) A^i \right). \tag{28}$$

Similarly, the gradient of $J(\theta)_{chunk}$ can be approximated by the gradient of following expression::

$$J(\theta)_{chunk} = s_1^1 - s_1^2, \tag{29}$$

where

$$
\begin{aligned}
s_1^i(\theta) &= \left( \prod_{t \in ch_1} \frac{p_\theta\left(x_{t-1}^i | x_t^i, c\right)}{p_{\theta_{old}}\left(x_{t-1}^i | x_t^i, c\right)} \right)^{\frac{1}{cs_1}} \\
&= \left( \prod_{t=1}^{T} \frac{p_\theta\left(x_{t-1}^i | x_t^i, c\right)}{p_{\theta_{old}}\left(x_{t-1}^i | x_t^i, c\right)} \right)^{\frac{1}{T}} \\
&= \left( \prod_{t=1}^{T} r_t^i(\theta) \right)^{\frac{1}{T}}, \quad i = 1, 2.
\end{aligned}
\tag{30}
$$

In Proximal Policy Optimization (PPO) (Schulman et al., 2017) and GRPO-based methods, the importance ratio $r_t^i(\theta)$ remains close to 1 due to trust-region constraints Schulman et al. (2015; 2017). We therefore set:

$$r_t^i(\theta) = 1 + \epsilon_t^i, \tag{31}$$

where $\epsilon_t^i$ is a minimal term. Substituting into Equation (24) and Equation (25):

$$J(\hat{\theta}) = \sum_{t \in T_a} \left( \epsilon_t^1 - \epsilon_t^2 \right) + \sum_{t \in T_{ia}} \left( \epsilon_t^2 - \epsilon_t^1 \right) \tag{32}$$

$$
\begin{aligned}
J(\theta)_{GRPO} &= \sum_{t \in T_a} \left( \epsilon_t^1 - \epsilon_t^2 \right) + \sum_{t \in T_{ia}} \left( \epsilon_t^1 - \epsilon_t^2 \right) \\
&= \sum_{t=1}^{T} \left( \epsilon_t^1 - \epsilon_t^2 \right).
\end{aligned}
\tag{33}
$$

For the chunk-level ratio in Equation (30), applying the logarithm and Taylor expansion gives:

$$
\begin{aligned}
s_1^i(\theta) &= \left(\prod_{t=1}^{T} r_t^i(\theta)\right)^{\frac{1}{T}} \\
&= \left(\prod_{t=1}^{T} \left(1 + \epsilon_t^i\right)\right)^{\frac{1}{T}} \\
&= 1 + \frac{1}{T} \sum_{1}^{T} \epsilon_t^i.
\end{aligned}
\tag{34}
$$

Thus the chunk-level objective reduces to:

$$
\begin{aligned}
J(\theta)_{chunk} &= s_1^1 - s_1^2 \\
&= \left(1 + \frac{1}{T} \sum_{1}^{T} \epsilon_t^1\right) - \left(1 + \frac{1}{T} \sum_{1}^{T} \epsilon_t^2\right) \\
&= \frac{1}{T} \sum_{t=1}^{T} \left(\epsilon_t^1 - \epsilon_t^2\right) \\
&= \frac{1}{T} J(\theta)_{GRPO}.
\end{aligned}
\tag{35}
$$

This shows that chunk-level optimization yields a smoothed version of the step-level GRPO objective. More formally, by comparing the squared distances between coefficient vector of $J(\hat{\theta})$, $J(\theta)_{GRPO}$, and $J(\theta)_{chunk}$, we find:

$$
\begin{aligned}
\|J(\hat{\theta}) - J(\theta)_{GRPO}\|_2^2 &= 2m \times (1 - (-1))^2 \\
&= 8m.
\end{aligned}
\tag{36}
$$

$$
\begin{aligned}
\|J(\hat{\theta}) - J(\theta)_{chunk}\|_2^2 &= \|J(\hat{\theta}) - \frac{1}{T} J(\theta)_{GRPO}\|_2^2 \\
&= \|J(\hat{\theta})\|^2 + \frac{1}{T^2}\|J(\theta)_{GRPO}\|^2 - \frac{2}{T} J(\hat{\theta}) \cdot J(\theta)_{GRPO} \\
&= 2T + \frac{2T}{T^2} - \frac{2}{T} \cdot 2\,(T - 2m) \\
&= 2T - 4 + \frac{8m + 2}{T},
\end{aligned}
\tag{37}
$$

Where $m$ denotes the number of inaccurately attributed timesteps, which we mentioned in the beginning of this section. We want Equation (37) to be smaller than Equation (36), i.e.,

$$
\|J(\hat{\theta}) - J(\theta)_{GRPO}\|_2^2 - \|J(\hat{\theta}) - J(\theta)_{chunk}\|_2^2 \geq 0
\tag{38}
$$

Solving yields:

$$
\begin{aligned}
&\|J(\hat{\theta}) - J(\theta)_{GRPO}\|_2^2 - \|J(\hat{\theta}) - J(\theta)_{chunk}\|_2^2 \geq 0 \\
\Leftrightarrow\,& 8m - 2T + 4 - \frac{8m + 2}{T} \geq 0 \\
\Leftrightarrow\,& 2T^2 - (4m + 8)T + (8m + 2) \leq 0 \\
\Leftrightarrow\,& T^2 - (2m + 4)T + (4m + 1) \leq 0 \\
\Leftrightarrow\,& m - \sqrt{m^2 + 3} + 2 \leq T \leq m + \sqrt{m^2 + 3} + 2
\end{aligned}
\tag{39}
$$

Since $1 \leq m \leq T$, the first inequality always holds. As both $T$ and $m$ are positive integers, we obtain:

$$T \begin{cases} \leq 5, & \text{if } m = 1 \\ \leq 2m + 2, & \text{if } m \geq 2. \end{cases} \tag{40}$$

Note that here $cs_1 = T$, and the whole trajectory is treated as a single chunk. When the chunk size $cs \leq 5$, Equation (38) always holds, meaning that the chunk-level objective $J(\theta)_{chunk}$ is closer to the ground-truth object $J(\hat{\theta})$ than $J(\theta)_{GRPO}$. For larger chunks, Equation (38) still holds when $m \leq \frac{T-2}{2}$.

The insights of this solution are:

- For small chunks (e.g. $cs_j = 5$), chunk-level optimization always outperforms step-level GRPO.
- For large chunk sizes, it also holds when roughly half of the timesteps suffer from inaccurate advantage attribution.
- From Equation (35), chunk-level optimization consistently provides smoother gradients than step-level GRPO..

Moreover, here we present another perspective demonstrating Chunk-GRPO's superiority based on the gradient weights, which shows why Chunk-GRPO is more stable.

To begin with, the gradient of the GRPO's objective in Equation (5) can be derived as follows (clipping and KL are omitted for brevity):

$$\nabla_\theta J_{GRPO}(\theta) = \nabla_\theta E_{c,x} \left[ \frac{1}{G} \frac{1}{T} \sum_{i=1}^{G} \sum_{t=1}^{T} r_t^i(\theta) A^i \right]$$

$$\tag{41}$$

$$= \nabla_\theta E_{c,x} \left[ \frac{1}{G} \sum_{i=1}^{G} A^i \frac{1}{T} \sum_{t=1}^{T} r_t^i(\theta) \cdot \nabla_\theta \log\left(r_t^i(\theta)\right) \right].$$

Given Equation (6), this simplifies to:

$$\nabla_\theta J_{GRPO}(\theta) = \nabla_\theta E_{c,x} \left[ \frac{1}{G} \sum_{i=1}^{G} A^i \frac{1}{T} \sum_{t=1}^{T} \frac{p_\theta(x_{t-1}^i|x_t^i,c)}{p_{\text{old}}(x_{t-1}^i|x_t^i,c)} \cdot \nabla_\theta \log\left(p_\theta(x_{t-1}^i|x_t^i,c)\right) \right]. \tag{42}$$

In comparison, the gradient of our Chunk-GRPO's objective in Equation (13) is derived below. For convenience, we denote the chunk-level importance ratio from Equation (14) as:

$$s_j^i(\theta) = \left( \prod_{t \in ch_j} \frac{p_\theta\left(x_{t-1}^i|x_t^i,c\right)}{p_{\theta_{old}}\left(x_{t-1}^i|x_t^i,c\right)} \right)^{\frac{1}{cs_j}}, \tag{43}$$

Then the gradient becomes to:

$$\nabla_\theta J_{Chunk-GRPO}(\theta) = \nabla_\theta E_{c,x} \left[ \frac{1}{G} \frac{1}{K} \sum_{i=1}^{G} \sum_{j=1}^{K} s_j^i(\theta) A^i \right]$$

$$= \nabla_\theta E_{c,x} \left[ \frac{1}{G} \sum_{i=1}^{G} A^i \frac{1}{K} \sum_{j=1}^{K} s_j^i(\theta) \cdot \nabla_\theta \log\left(s_j^i(\theta)\right) \right]$$

$$= \nabla_\theta E_{c,x} \frac{1}{G} \sum_{i=1}^{G} A^i$$

$$\left[ \frac{1}{K} \sum_{j=1}^{K} \left( \prod_{t \in ch_j} \frac{p_\theta\left(x_{t-1}^i|x_t^i,c\right)}{p_{\theta_{old}}\left(x_{t-1}^i|x_t^i,c\right)} \right)^{\frac{1}{cs_j}} \cdot \frac{1}{cs_j} \sum_{t \in ch_j} \nabla_\theta \log\left(p_\theta(x_{t-1}^i|x_t^i,c)\right) \right]$$

$$\tag{44}$$

Therefore, the fundamental distinction between ours and GRPO lies in how they weight the gradients of the log likelihoods of tokens. In standard GRPO, tokens are weighted individually according to their respective importance weight $\frac{p_\theta(x_{t-1}^i|x_t^i,c)}{p_{\text{old}}(x_{t-1}^i|x_t^i,c)}$. However, these unequal weights, which can vary among $(0, 1+\epsilon)$ for $A \geq 0$ or $[1-\epsilon, \infty)$ for $A \leq 0$, are not negligible, and their impact can accumulate and lead to unstable consequences. In contrast, our approach applies a unified weight $\left(\prod_{t\in ch_j} \frac{p_\theta(x_{t-1}^i|x_t^i,c)}{p_{\theta_{old}}(x_{t-1}^i|x_t^i,c)}\right)^{\frac{1}{cs_j}}$ to all tokens within a chunk, effectively smoothing these fluctuations and eliminating this instability.

## B    RELATED WORK

### B.1    REINFORCEMENT LEARNING FOR DIFFUSION-BASED IMAGE GENERATION

Diffusion models (Ho et al., 2020; Rombach et al., 2022; Podell et al., 2023; Labs et al., 2025; Wu et al., 2025) have become one of the dominant paradigms for T2I generation. Early works (Xu et al., 2023; Black et al., 2024b; Fan et al., 2023) introduced RL into diffusion models through policy gradient optimization. Preference-based methods (Wallace et al., 2024; Sun et al., 2025a;c;d;e) were later developed, achieving competitive alignment without explicit reward modeling.

More recently, GRPO (Shao et al., 2024; Sun et al., 2025b) has attracted attention as an efficient alternative. Dance-GRPO (Xue et al., 2025) and Flow-GRPO (Liu et al., 2025b) pioneered the use of GRPO for T2I generation, unifying diffusion and flow matching through an SDE-based reformulation. MixGRPO (Li et al., 2025a) further improved efficiency via a mixed ODE–SDE paradigm. TempFlow-GRPO (He et al., 2025) introduced temporal-aware weighting across denoising steps. Pref-GRPO (Wang et al., 2025) identified the issue of illusory advantage and reformulated the optimization objective as pairwise preference fitting. BranchGRPO (Li et al., 2025c) restructured the rollout process into a branching tree, amortizing computation across shared prefixes.

In contrast to these works, our approach explicitly addresses two key issues in GRPO-based T2I generation: (1) inaccurate advantage attribution, and (2) neglect of temporal dynamics. By introducing chunk-level optimization guided by the inherent temporal structure of flow matching, we enhance GRPO from the perspective of optimization granularity.

## C    EXPERIMENT DETAILS

### C.1    CHUNK CONFIGURATION

In practice, the default Chunk-GRPO segments the image generation trajectory into $K = 4$ chunks with $cs_j\,{}_{j=1}^4 = 2, 3, 4, 7$ under $T = 17$ [4] timesteps. The rationale is as follows:

- Following Figure 3, We set the first chunk as $cs_1 = 2$.
- For the last chunk, we first conduct a pre-observation: we compute the relative $L1$ distance in Equation (15) again, but with a Dance-GRPO-trained model instead of the base model. As shown in Figure 8, RL alters the relative $L1$ distance primarily in the latter half of timesteps. Based on this, we set $ch_4 = 7$.
- For $ch_2 = 3$ and $ch_3 = 4$, we base the segmentation on the second derivative of the $L1$ curve.
- This configuration also satisfies the requirement in Proposition 1, which recommends keeping chunk size small (e.g. 5).

We emphasize that this segmentation is not guaranteed to be the only optimal choice. Exploring adaptive chunk configurations under different $T$ is an interesting direction for future work.

---

[4]We neglect the last timestep following Dance-GRPO, as the last step does not introduce stochasticity.

## C.2 Training Details

All experiments were conducted on 8 Nvidia H800 GPUs. The hyperparameters are summarized in Table 6.

## C.3 Evaluation Details

We set $T = 50$ during evaluation. Following (Li et al., 2025a), the first 30 steps are sampled with the trained model, while the remaining 20 steps are sampled with the base model. This hybrid inference strategy and corresponding settings, also used in (Li et al., 2025a), have proven effective in mitigating reward hacking.

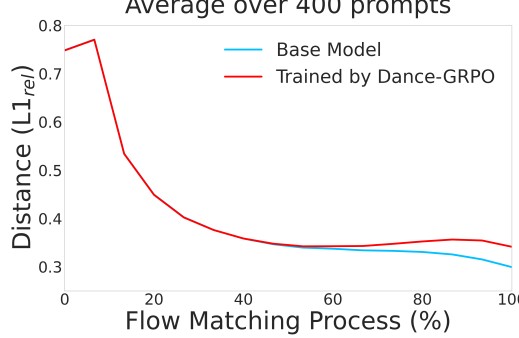

Figure 8: The relative $L1$ distance comparison, before and after the training of Dance-GRPO.

Table 6: Hyperparameter Settings

| Parameter | Value | Parameter | Value |
|---|---|---|---|
| Learning rate | $1 \times 10^{-5}$ | Weight decay | $1 \times 10^{-4}$ |
| Train batch size | 2 | SP size | 1 |
| SP batch size | 2 | Max grad norm | 0.01 |
| Resolution | $720 \times 720$ | Sampling steps | 17 |
| Eta | 0.7 | Num. generations | 12 |
| Grad. accum. steps | 12 | Shift (branch offset) | 3 |
| Clip range | $5 \times 10^{-5}$ | Training steps | 150 |
| Timestep fraction | 0.5 | | |

## C.4 Addition baseline comparisons

we conducted an additional comparison with Pref-GRPO(Wang et al., 2025) on the preference alignment tasks. As shown in Table 7, our approach consistently outperforms baselines like Dance-GRPO and Pref-GRPO, confirming the effectiveness of our chunk-level optimization.

Table 7: Results on Preference Alignment

| Model | HPSv3 | ImageReward |
|---|---|---|
| Flux | 13.804 | 1.086 |
| Pref-GRPO | 14.868 | 1.139 |
| Chunk-GRPO w/o ws | 15.236 | 1.147 |
| Chunk-GRPO w/ ws | 15.373 | 1.149 |

[1] The 'ws' refers to the weighted sampling strategy.

## C.5 Adaptive Chunking

While our initial implementation used fixed chunks based on the $L1$ dynamics shown in Figure 3 has demonstrated significant results in experiments, it is sensitive to the sampling steps and lacks adaptivity. To address this issue, we have developed an adaptive chunking strategy as follows.

Given a total sampling step $T$ and a prompt $c$, we can obtain the relative

Table 8: Results of Adaptive Chunking

| Model | HPSv3 | ImageReward |
|---|---|---|
| Flux | 13.804 | 1.086 |
| Dance-GRPO | 15.080 | 1.141 |
| Chunk-GRPO w/o ac | 15.236 | 1.147 |
| Chunk-GRPO w/ ac | 15.213 | 1.147 |

[1] The 'ac' refers to the adaptive chunking.

$L1$ curve of the generation trajectory. First, we compute its first-order derivatives for each timestep, and group consecutive steps with the same sign into chunks. If the derivative signs are uniform across the entire trajectory, we split the sequence at the midpoint. Then, we recursively apply the above to each resulting chunk, but using second-order derivatives, third-order derivatives, and so on. When a chunk is sufficiently small ($cs_j \leq 5$), then it gets stopped from recursion. When all chunks get stopped, then the chunking is finished.

Table 8 presents the results of this adaptive chunking. Without any hyperparameters fine-tuning, the adaptive chunking achieves remarkable results, which reveals the potential of adaptive chunking robust to prompts and sampling timesteps, further highlighting the superiority of our chunk-level optimization.

### C.6 USER STUDIES

A user study is the most robust validation for alignment methods. Since our task focuses heavily on preference alignment, we conducted a user study to assess human preferences.

During the test phase, ten reviewers possessing good aesthetic sensibilities were recruited and administered a qualifying assessment, in which nine participants successfully passed. We extracted 40 prompts from our evaluation dataset - HPDv2.1 test set(Wu et al., 2023). Reviewers were tasked with identifying the optimal image based on the aesthetics from a set of 3 alternatives generated by: Dance-GRPO, Chunk-GRPO without weighted sampling (ws), and Chunk-GRPO with ws.

The results in Table 9 show that the combined Chunk-GRPO variants were preferred by human reviewers 72.5% of the time, which further demonstrates the significant superiority of our approach in human preference alignment, validating its effectiveness.

Table 9: Results of User Studies

| Method | Win Rate |
|---|---|
| Dance-GRPO | 0.275 |
| Chunk-GRPO w/o ws | 0.350 |
| Chunk-GRPO w/ ws | 0.375 |

[1] The 'ws' refers to the weighted sampling strategy.

### D REPRODUCIBILITY STATEMENT

We have made every effort to ensure the reproducibility of our results. The source code is provided in the 'Supplementary Material', and training details are described in Section 5.1 and Section C.

### E ADDITIONAL VISUALIZATION

Figure 9 and 10 present qualitative comparisons among FLUX, Dance-GRPO, Chunk-GRPO without temporal dynamics, Chunk-GRPO with temporal dynamics and Chunk-GRPO with weighted sampling. Overall, Chunk-GRPO generates outputs that align more closely with human aesthetic preferences, exhibiting stronger lighting contrast, more vivid colors, and finer details.

2B from NieR Automata eating a bagel.

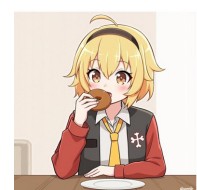 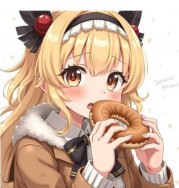 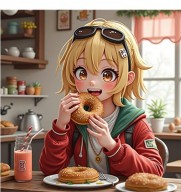 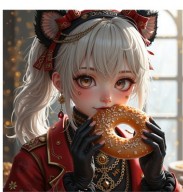 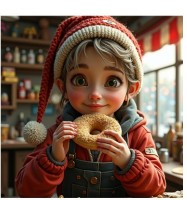

FLUX          DanceGRPO          Chunk-GRPO w/o temporal dynamics          Chunk-GRPO w/ temporal dynamics          Chunk-GRPO w/ weighted sampling

A girl with pink pigtails and face tattoos.

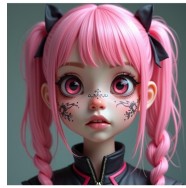 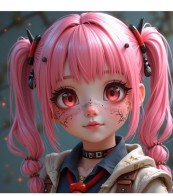 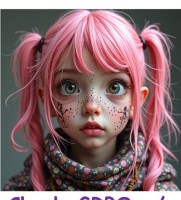 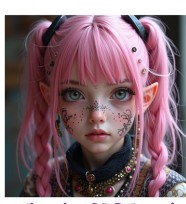 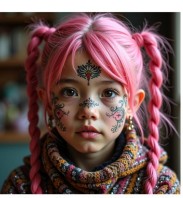

FLUX          DanceGRPO          Chunk-GRPO w/o temporal dynamics          Chunk-GRPO w/ temporal dynamics          Chunk-GRPO w/ weighted sampling

16-year-old teenager wearing a white bear-ear hat with a smirk on their face.

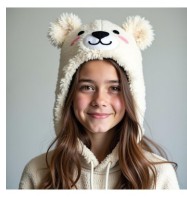 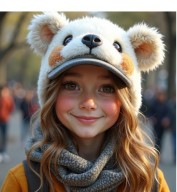 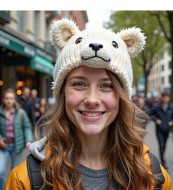 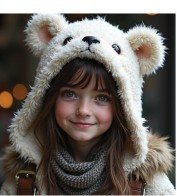 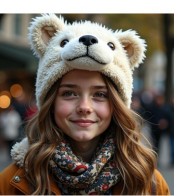

FLUX          DanceGRPO          Chunk-GRPO w/o temporal dynamics          Chunk-GRPO w/ temporal dynamics          Chunk-GRPO w/ weighted sampling

"The image is a Roy Lichtenstein emo portraying a woman with dark brown pixie hair, entirely black eyes, wearing a black tank top, leather jacket, skirt, choker, and boots."

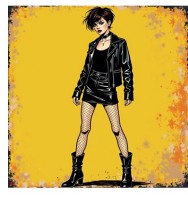 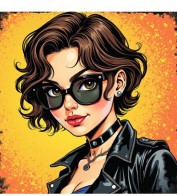 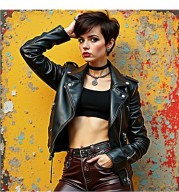 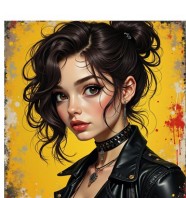 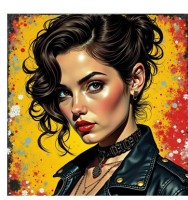

FLUX          DanceGRPO          Chunk-GRPO w/o temporal dynamics          Chunk-GRPO w/ temporal dynamics          Chunk-GRPO w/ weighted sampling

Figure 9: Additional visualization comparison between the FLUX, DanceGRPO, Chunk-GRPO w/o temporal dynamics, Chunk-GRPO w/ temporal dynamics and Chunk-GRPO w/ weighted sampling.

A corgi puppy with many eyes depicted in a horror manga drawn by Junji Ito.

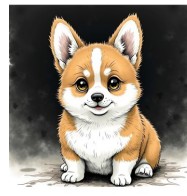 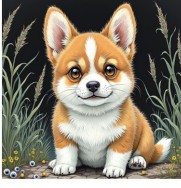 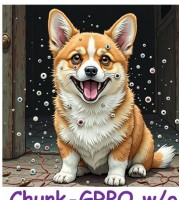 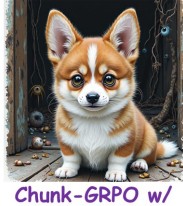 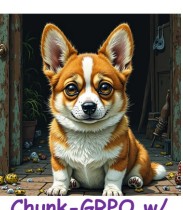

FLUX    DanceGRPO    Chunk-GRPO w/o temporal dynamics    Chunk-GRPO w/ temporal dynamics    Chunk-GRPO w/ weighted sampling

A young woman witch cosplaying with a magic wand and broom, wearing boots, and posing in a full body shot with a detailed face.

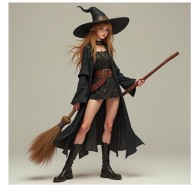 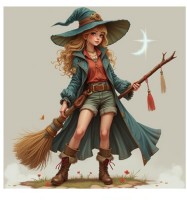 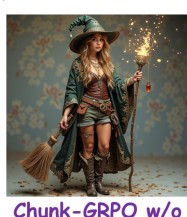 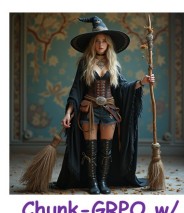 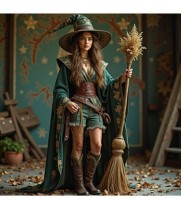

FLUX    DanceGRPO    Chunk-GRPO w/o temporal dynamics    Chunk-GRPO w/ temporal dynamics    Chunk-GRPO w/ weighted sampling

The image depicts a stunning supernova within a fantasy artwork on Artstation.

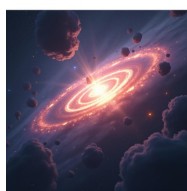 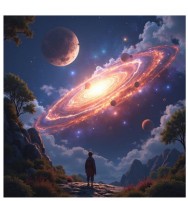 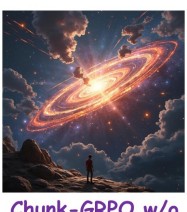 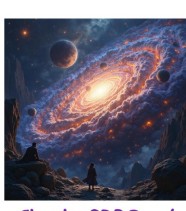 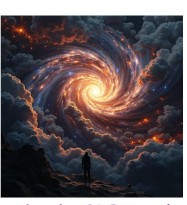

FLUX    DanceGRPO    Chunk-GRPO w/o temporal dynamics    Chunk-GRPO w/ temporal dynamics    Chunk-GRPO w/ weighted sampling

A train is moving along the track in the countryside.

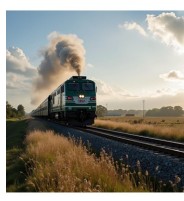 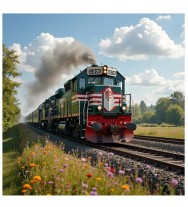 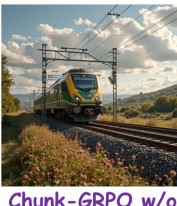 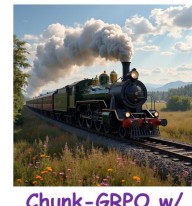 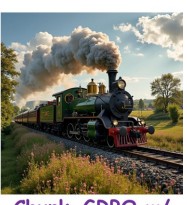

FLUX    DanceGRPO    Chunk-GRPO w/o temporal dynamics    Chunk-GRPO w/ temporal dynamics    Chunk-GRPO w/ weighted sampling

Figure 10: Additional visualization comparison between the FLUX, DanceGRPO, Chunk-GRPO w/o temporal dynamics, Chunk-GRPO w/ temporal dynamics and Chunk-GRPO w/ weighted sampling.

