# OpenReview forum: "Sample by Step, Optimize by Chunk: Chunk-Level GRPO for Text-to-Image Generation"
_ICLR.cc/2026/Conference — Submitted to ICLR 2026_

### Official Review · Reviewer_xpH9 · 2025-10-30

**Soundness:** 4
**Presentation:** 4
**Contribution:** 3
**Rating:** 6
**Confidence:** 4

**Summary:**

The paper introduces Chunk-GRPO, a novel reinforcement learning optimization paradigm for flow-matching-based text-to-image (T2I) generation. The method extends the Group Relative Policy Optimization (GRPO) by addressing two of its core limitations: (1) inaccurate advantage attribution across timesteps and (2) the neglect of temporal dynamics.

Instead of applying optimization at every single generation step, the proposed method groups consecutive timesteps into temporally coherent chunks and performs optimization at the chunk level. The paper also proposes an optional weighted sampling strategy, which biases optimization toward chunks that correspond to higher-noise regions of the generation trajectory.

The authors present both theoretical justification and comprehensive experiments demonstrating improvements in preference alignment and image quality over prior methods such as Dance-GRPO.

**Strengths:**

1. The paper is well-organized and easy to follow.
2. Chunk-GRPO is conceptually straightforward, requires only minor modifications to existing GRPO frameworks, and can be easily implemented.
3. The paper includes a clean mathematical analysis showing why chunk-level optimization yields smoother gradients and more accurate updates under imperfect advantage attribution.
4. The analysis of chunk sizes and the discussion of how temporal dynamics guide segmentation provide clear practical guidance for choosing chunk configurations in future applications.
5. The authors perform extensive experiments across multiple datasets and reward models, demonstrating consistent gains in preference alignment and image fidelity.

**Weaknesses:**

1. Missing significance analysis. Reported metrics are presented without standard deviations. Without these, it is difficult to assess whether the observed improvements are statistically significant.
2. A user study is missing. Since the paper focuses heavily on preference alignment and perceptual image quality, a user study would provide validation of the improvements.
3. The improvement is rather small, especially considering the results on WISE.

**Questions:**

1. Have you experimented with adaptive chunking, where the chunk boundaries evolve during training?

---

> ### Author Response · Authors · 2025-11-28
> **Part 1/2**
>
> Dear Reviewer xpH9,
>
> Thank you very much for your comprehensive and detailed review of our paper. We greatly appreciate your recognition of our work, including the straightforward concept, solid analysis, and extensive pratice. Your suggestions, particularly regarding adaptive chunking, have been invaluable. Please find our detailed responses below.
>
> # W1: Significance Analysis
> > Reported metrics are presented without standard deviations $\cdots$ statistically significant.
>
> Thank you for this question. Actually our experiment was conducted with 3 random seeds:
>
> **Table 10.** Results on Preference Alignment.
> | Model | HPSv3 | ImageReward |
> | ----- | ----------- | -------- |
> | Flux | 13.804 | 1.086 |
> | Dance-GRPO | 15.080 $\pm$ 0.058 | 1.141 $\pm$ 0.003|
> | Chunk-GRPO w/o ws | **15.236** $\pm$ 0.021| **1.147** $\pm$ 0.002|
>
> These results demonstrate the stability and statistical significance of our method's improvement. The reason why we did not include the standard deviations in the metric was that it was the routine presentation style of many Text-to-Image (T2I) papers[1,2,3].
>
> # W2: User Study
> > A user study is missing $\cdots$ a user study would provide validation of the improvements.
>
> Thank you for this valuable suggestion. We agree completely that a user study is the most robust validation for alignment methods. Following your suggestion, we conducted a study to assess human preferences.
>
> During the test phase, ten reviewers possessing good aesthetic sensibilities were recruited and administered a qualifying assessment, in which nine participants successfully passed. We extracted 40 prompts from our evaluation dataset - HPDv2.1 test set[4]. Reviewers were tasked with identifying the optimal image based on the aesthetics from a set of 3 alternatives generated by: Dance-GRPO, Chunk-GRPO without weighted sampling (ws), and Chunk-GRPO with ws.
>
> **Table 9.** Results of User Studies.
> | Method | Win Rate |
> | ----- | ----------- |
> | Dance-GRPO | 0.275 |
> | Chunk-GRPO w/o ws | 0.350|
> | Chunk-GRPO w/o ws | **0.375**|
>
> The results in Table 9 show that the combined Chunk-GRPO variants were preferred by human reviewers $72.5$% of the time, which further demonstrates the significant superiority of our approach in human preference alignment, validating its effectiveness.
>
> We have incorperated these into the paper.
>
> # W3: Small Improvement
> > The improvement is rather small, especially considering the results on WISE.
>
> Thank you for pointing it out. In our original report, Chunk-GRPO demonstrates significant improvements in preference alignment, while moderate gains on the text-to-image (T2I) benchmark, WISE.
>
> To further validate Chunk-GRPO's effectiveness, we conducted experiments on the standard T2I benchmark GenEval[2], using CLIP[3] as the reward model.
>
> **Table 5.** Results on GenEval.
> | Model | Single Obj. | Two Obj. | Counting | Colors | Position | Color Attri. | Overall |
> | ----- | ----------- | -------- | -------- | ------ | -------- | ------------ | ------- |
> | Flux | 0.99 | 0.83 | 0.71 | 0.75 | **0.24** | 0.44 | 0.66 |
> | Dance-GRPO | **1.00** | **0.86** | 0.71 | 0.78 | 0.22 | 0.46 | 0.67 |
> | Chunk-GRPO w/o ws | 0.99 | 0.85 | **0.75** | **0.81** | 0.21 | **0.51** | **0.69** |
>
> Results in Table 5 demonstrate that Chunk-GRPO outperforms the standard step-level GRPO (Dance-GRPO) significantly. Specifically, Chunk-GRPO achieves a performance gain of 0.03, which is $3$ times larger than the gain achieved by Dance-GRPO.
>
> We attribute the moderate improvement on WISE to a misalignment between the evaluation task and the reward model. Specifically, WISE focuses on reasoning capabilities in T2I generation, whereas HPSv3 is a reward model optimized primarily for aesthetic alignment. As a proof, the results in Table 1 demonstrate the significant outperformance of our method in aesthetic preference alignment.
>
> We have incorperated these into the paper.

---

> ### Author Response · Authors · 2025-11-28
> **Part 2/2**
>
> # Q1: Adaptive Chunking
> > Have you experimented with adaptive chunking, where the chunk boundaries evolve during training?
>
> Thank you for your insightful comment. While our initial implementation used fixed chunks, we have developed an adaptive chunking strategy as follows to address your concerns regarding sensitivity to sampling steps and prompts.
>
> Given a total sampling step $T$ and a prompt $c$, we can obtain the relative $L1$ curve of the generation trajectory. First, we compute its first-order derivatives for each timestep, and group consecutive steps with the same sign into chunks. If the derivative signs are uniform across the entire trajectory, we split the sequence at the midpoint. Then, we recursively apply above to each resulting chunks, but using second-order derivatives, third-order derivatives, and so on. When a chunk is sufficiently small ($cs_j \leq 5$), then this chunk gets stopped from recursion. When all chunks get stopped, then the chunking is done.
>
> **Table 8.** Results of Adaptive Chunking (ac).
> | Model | HPSv3 | ImageReward |
> | ----- | ----------- | -------- |
> | Flux | 13.804 | 1.086 |
> | Dance-GRPO | 15.080 | 1.141 |
> | Chunk-GRPO w/o ac | **15.236** | **1.147** |
> | Chunk-GRPO w/ ac | 15.213 | **1.147** |
>
> Table 8 presents the results of this adaptive chunking. Without any hyperparameters fine-tuning, the adaptive chunking achieves remarkable results, which reveals the potential of adaptive chunking robust to prompts and sampling timesteps, further highlighting the superiority of our chunk-level optimization.
>
> We will do more exploration following this direction and incorperate it into the paper. Many thanks for your suggestions!
>
> Thank you once again for your time, support, and constructive feedbackthroughout the review process. Your feedback and engagement, especially the adaptive chunking, have been truly insightful and have helped us improve our work significantly. We promise that we will do more exploration following this direction and incorperate it into our paper.
>
>
> # References
> [1] Liu, J., Liu, G., Liang, J., Li, Y., Liu, J., Wang, X., Wan, P., Zhang, D. and Ouyang, W., 2025. Flow-grpo: Training flow matching models via online rl. arXiv preprint arXiv:2505.05470.
>
> [2] Li, J., Cui, Y., Huang, T., Ma, Y., Fan, C., Yang, M. and Zhong, Z., 2025. Mixgrpo: Unlocking flow-based grpo efficiency with mixed ode-sde. arXiv preprint arXiv:2507.21802
>
> [3] Li, Y., Wang, Y., Zhu, Y., Zhao, Z., Lu, M., She, Q. and Zhang, S., 2025. Branchgrpo: Stable and efficient grpo with structured branching in diffusion models. arXiv preprint arXiv:2509.06040.
>
> [4] Wu, X., Hao, Y., Sun, K., Chen, Y., Zhu, F., Zhao, R. and Li, H., 2023. Human preference score v2: A solid benchmark for evaluating human preferences of text-to-image synthesis. arXiv preprint arXiv:2306.09341.

---

> ### Author Response · Authors · 2025-12-03
> **Update on the Adaptive Chunking**
>
> We have incorperated the adaptive chunking into the Appendix C.

---

### Official Review · Reviewer_k7kg · 2025-11-01

**Soundness:** 2
**Presentation:** 3
**Contribution:** 2
**Rating:** 4
**Confidence:** 4

**Summary:**

This paper argues that existing reinforcement learning methods for text-to-image generation place excessive emphasis on global advantages while neglecting local advantages, which may prevent trajectories from converging to the optimal solution. The authors propose a chunk-GRPO approach that segments the generated trajectory into chunks and computes advantages for each segment to address this issue. In addition, temporal dynamics is incorporated to dynamically adjust chunk sizes. Experimental results demonstrate improved performance on text-to-image generation tasks.

**Strengths:**

(1)	The authors propose the idea of incorporating both global and local advantages to evaluate the optimality of trajectory sequences, which is an interesting direction worthy of further exploration.
(2)	The proposed temporal dynamics method avoids complex hyperparameter configurations, thereby enhancing the generality of the approach.

**Weaknesses:**

(1)	More intuitive results: The example provided in Figure 2 of the paper is merely a schematic illustration. The authors are encouraged to present real image cases demonstrating whether, during the early or middle stages of generation, intermediate images exhibit higher quality, yet the final convergence results in an inferior output.
(2)	Motivation concern: Although the authors argue that certain steps in the generation process may possess local advantages, I believe that a well-formed generation trajectory does not—and need not—ensure local optimality at every step. The objective of reinforcement learning should still focus on achieving global optimality.
(3)	Experimental concerns: Based on the experimental results presented in the tables, the performance gains from chunking are minimal. This further calls into question the necessity of the chunking operation.

**Questions:**

All of my concerns are presented in the Weaknesses section.

---

> ### Author Response · Authors · 2025-11-28
> **Part 1/2**
>
> Dear Reviewer k7kg,
>
> Thank you for your detailed review of our paper. We greatly appreciate your recognition of the novelty and generality. Please find our detailed responses to your concerns below.
>
> # W1: Intuitive Results
> > The example provided in Figure 2 $\cdots$ intermediate images exhibit higher quality, yet the final convergence results in an inferior output.
>
> Thank you for your careful review. We want to clarify the interpretation of Figure 3 (Figure 2 in the original version).
>
> The Y-axis in Figure 3 does not represent image quality. Instead, it measures the relative $L1$ distance in Equation (15):
> $$L1_\text{rel} (x, t) = \frac{\lVert x_t - x_{t-1} \rVert_1}{\lVert x_t \rVert_1}.$$
> Therefore, the curves in Figure 3 illustrate the magnitude of change experienced by the intermediate images in the latent space, not their quality. Here we attached the corresponding average $L1$ statistics over $400$ prompts (the right subfigure in Figure 3):
> $$[0.748, 0.771, 0.534, 0.445, 0.403, 0.376, 0.359, 0.347, 0.340, 0.337, 0.334, 0.333, 0.331, 0.326, 0.315, 0.299].$$
> Therefore, the figure indicates that the intermediate images in the early stage exhibit larger changes in the latent space.
>
> # W2: Motivation Concern
> > Although the authors argue that certain steps in the generation process may possess local advantages $\cdots$ The objective of reinforcement learning should still focus on achieving global optimality.
>
> Thank you for pointing out this potential confusion. We want to clarify that we do not compute or assign local advantages. Instead, we still only use the final outcome reward (final image reward) as the sole basis for the advantage term, exactly as what standard GRPO does. Our novelty lies in that we shift the optimization from step-level to chunk-level. Our idea, as well as our pratice, fully stands with your assertion "I believe that a well-formed generation trajectory does not — and need not — ensure local optimality at every step".
>
> To be specific, by "possess local advantages", you may mean the incorrect advantage attribution we pointed out. If so, there is a misunderstanding. While we attempt to mitigate the incorrect advantage attribution issue, instead of computing local advantages, we achieve this by changing only the importance ratio to chunk-level. As shown in Equation (14), the chunk-level optimization objective is:
> $$\begin{aligned}
>     J(\theta) &= E_{c, x} \\\\\\
>     &\left[\frac{1}{G} \frac{1}{K} \sum_{i=1}^{G} \sum_{j=1}^{K}\left(min \left(r_j^i \left(\theta \right)A^i, clip\left(r_j^i \left(\theta \right), 1- \epsilon, 1 + \epsilon \right)A^i\right) - \beta D_{KL}\left(\pi_\theta || \pi_{ref}\right) \right)\right].
> \end{aligned}$$
> Crucially, the $A^i$ keeps the same for every chunk $ch_j$, using the final outcome reward as the only reward, as what GRPO does. Our change is confined to the chunk-level importance ratio, which shifts from the orginal GRPO's in Equation (10) to Equation (14):
> $$r_j^i(\theta) = \left( \prod_{t \in ch_j} \frac{p_\theta \left( x_{t-1}^i | x_t^i, c \right)}{p_{\text{old}} \left( x_{t-1}^i | x_t^i, c \right)} \right) ^{\frac{1}{cs_j}}.$$
> Therefore, we do not assign local advantages or pursue local optimality. Instead, we still obtain the final outcome reward as the only reward and assign it to each chunk. However, we shift the importance ratio, as well as the optimization, from step-level to chunk-level, which makes the optimization more stable, which is demonstrated in Appendix A.

---

> ### Author Response · Authors · 2025-11-28
> **Part 2/2**
>
> # W3: Experimental Concerns
> > Based on the experimental results $\cdots$ This further calls into question the necessity of the chunking operation..
>
> Thank you for pointing it out. In our original report, Chunk-GRPO demonstrates significant improvements in preference alignment, while moderate gains on the text-to-image (T2I) benchmark, WISE.
>
> To further validate Chunk-GRPO's effectiveness, we conducted experiments on the standard T2I benchmark GenEval[2], using CLIP[3] as the reward model.
>
> **Table 5.** Results on GenEval.
> | Model | Single Obj. | Two Obj. | Counting | Colors | Position | Color Attri. | Overall |
> | ----- | ----------- | -------- | -------- | ------ | -------- | ------------ | ------- |
> | Flux | 0.99 | 0.83 | 0.71 | 0.75 | **0.24** | 0.44 | 0.66 |
> | Dance-GRPO | **1.00** | **0.86** | 0.71 | 0.78 | 0.22 | 0.46 | 0.67 |
> | Chunk-GRPO w/o ws | 0.99 | 0.85 | **0.75** | **0.81** | 0.21 | **0.51** | **0.69** |
>
> Results in Table 5 demonstrate that Chunk-GRPO outperforms the standard step-level GRPO (Dance-GRPO) significantly. Specifically, Chunk-GRPO achieves a performance gain of 0.03, which is $3$ times larger than the gain achieved by Dance-GRPO.
>
> We attribute the moderate improvement on WISE to a misalignment between the evaluation task and the reward model. Specifically, WISE focuses on reasoning capabilities in T2I generation, whereas HPSv3 is a reward model optimized primarily for aesthetic alignment. As a proof, the results in Table 1 demonstrate the significant outperformance of our method in aesthetic preference alignment.
>
> We have incorperated these into the paper.
>
> Thank you once again for your response, your engagement, and your commitment to helping improve our paper. We truly appreciate the time and effort you have dedicated to reviewing our work.
>
> # References
> [1] Ghosh, Dhruba, Hannaneh Hajishirzi, and Ludwig Schmidt. "Geneval: An object-focused framework for evaluating text-to-image alignment." Advances in Neural Information Processing Systems 36 (2023): 52132-52152.
>
> [2] Radford, A., Kim, J.W., Hallacy, C., Ramesh, A., Goh, G., Agarwal, S., Sastry, G., Askell, A., Mishkin, P., Clark, J. and Krueger, G., 2021, July. Learning transferable visual models from natural language supervision. In International conference on machine learning (pp. 8748-8763). PmLR.

---

### Official Review · Reviewer_VUdY · 2025-11-01

**Soundness:** 3
**Presentation:** 3
**Contribution:** 3
**Rating:** 4
**Confidence:** 4

**Summary:**

The paper proposes Chunk-Level GRPO, which shifts GRPO optimization from step-level to chunk-level for flow-matching text-to-image models to address inaccurate advantage attribution and the neglect of temporal dynamics during generation.​ Consecutive timesteps are grouped into chunks guided by prompt-invariant relative L1 latent dynamics, and policies are optimized using a chunk-level importance ratio, with an optional weighted sampling strategy that emphasizes high-noise chunks.​ Experiments on FLUX Dev with HPDv2.1 show consistent gains in preference alignment (HPSv3, ImageReward) and competitive WISE benchmark results, alongside ablations on chunk configurations, per-chunk training, and reward-model robustness, plus analysis of a stability trade-off introduced by weighted sampling.

**Strengths:**

- The motivation claimed in Figure 1 is very interesting and insightful. Additionally, the findings on temporal dynamics are beneficial to the community.

- Chunk boundaries are informed by prompt-invariant temporal dynamics via relative L1 distance, yielding a principled, dynamics-aware segmentation rather than arbitrary chunking.

**Weaknesses:**

I thank the authors for their efforts in this work. Below are some concerns about this paper.
- This work claims to be the first “chunk‑level” method but does not compare against other GRPO variants like Flow-GRPO, Pref-GRPO, it cited, weakening the contribution boundary beyond a single Dance‑GRPO baseline. Moreover, the proposed method performs only on par with Dance‑GRPO on WISE.

- The chunking implementation is heuristic. Boundaries are precomputed from relative L1 latent dynamics and kept fixed, lacking adaptivity and making performance sensitive to the sampling step $T$, the model, and certain prompts.

- Despite the insight the authors claimed in Figure 1, such a chunk-based design did not show an optimal approach to solving such issues. In some cases, there are still issues, such as $ Chunk_{1}$ has the greater final reward (advantage), its $t=1$ timestep is worse
than that in $Chunk_{2}$.

**Questions:**

Please see the #Weakness part.

---

> ### Author Response · Authors · 2025-11-28
> **Part 1/2**
>
> Dear Reviewer VUdY,
>
> Thank you very much for your comprehensive and detailed review of our paper. We are encouraged by your recognition of our insightful motivation and principled approachas. Your feedback, particularly regarding adaptive chunking, has been invaluable. Please find our point-by-point responses below.
>
> # W1: Baselines
> > This work claims to be the first "chunk‑level" $\cdots$ performs only on par with Dance‑GRPO on WISE.
>
> Thank you for this suggestion. First, we have added a comparison with Pref-GRPO[1] on the preference alignment tasks. As shown in Table 7, our approach consistently outperforms baselines like Dance-GRPO and Pref-GRPO, confirming the effectiveness of our chunk-level optimization.
>
> **Table 7.** Results on Preference Alignment.
> | Model | HPSv3 | ImageReward |
> | ----- | ----------- | -------- |
> | Flux | 13.804 | 1.086 |
> | Dance-GRPO | 15.080 | 1.141 |
> | Pref-GRPO | 14.868 | 1.139 |
> | Chunk-GRPO w/o ws | 15.236 | 1.147 |
> | Chunk-GRPO w/ ws | **15.373** | **1.149** |
>
> Our approach consistently outperforms both baselines like Dance-GRPO and Pref-GRPO, confirming the effectiveness of chunk-level optimization.
>
> Second, in our original report, Chunk-GRPO demonstrates significant improvements in preference alignment, while moderate gains on the text-to-image (T2I) benchmark, WISE.
>
> To further validate Chunk-GRPO's effectiveness, we conducted experiments on the standard T2I benchmark GenEval[2], using CLIP[3] as the reward model.
>
> **Table 5.** Results on GenEval.
> | Model | Single Obj. | Two Obj. | Counting | Colors | Position | Color Attri. | Overall |
> | ----- | ----------- | -------- | -------- | ------ | -------- | ------------ | ------- |
> | Flux | 0.99 | 0.83 | 0.71 | 0.75 | **0.24** | 0.44 | 0.66 |
> | Dance-GRPO | **1.00** | **0.86** | 0.71 | 0.78 | 0.22 | 0.46 | 0.67 |
> | Chunk-GRPO w/o ws | 0.99 | 0.85 | **0.75** | **0.81** | 0.21 | **0.51** | **0.69** |
>
> Results in Table 5 demonstrate that Chunk-GRPO outperforms the standard step-level GRPO (Dance-GRPO) significantly. Specifically, Chunk-GRPO achieves a performance gain of 0.03, which is $3$ times larger than the gain achieved by Dance-GRPO.
>
> We attribute the moderate improvement on WISE to a misalignment between the evaluation task and the reward model. Specifically, WISE focuses on reasoning capabilities in T2I generation, whereas HPSv3 is a reward model optimized primarily for aesthetic alignment. As a proof, the results in Table 1 demonstrate the significant outperformance of our method in aesthetic preference alignment.
>
> We have incorperated these into the paper.
>
> # W2: Heuristic Chunking Implementation
> > The chunking implementation is heuristic $\cdots$ sensitive to the sampling step $T$, the model, and certain prompts.
>
> Thank you for this insightful comment. While our initial implementation used fixed chunks, we have developed an adaptive chunking strategy as follows to address your concerns regarding sensitivity to sampling steps and prompts.
>
> Given a total sampling step $T$ and a prompt $c$, we can obtain the relative $L1$ curve of the generation trajectory. First, we compute its first-order derivatives for each timestep, and group consecutive steps with the same sign into chunks. If the derivative signs are uniform across the entire trajectory, we split the sequence at the midpoint. Then, we recursively apply above to each resulting chunks, but using second-order derivatives, third-order derivatives, and so on. When a chunk is sufficiently small ($cs_j \leq 5$), then this chunk gets stopped from recursion. When all chunks get stopped, then the chunking is done.
>
> **Table 8.** Results of Adaptive Chunking (ac).
> | Model | HPSv3 | ImageReward |
> | ----- | ----------- | -------- |
> | Flux | 13.804 | 1.086 |
> | Dance-GRPO | 15.080 | 1.141 |
> | Chunk-GRPO w/o ac | **15.236** | **1.147** |
> | Chunk-GRPO w/ ac | 15.213 | **1.147** |
>
> Table 8 presents the results of this adaptive chunking. Without any hyperparameters fine-tuning, the adaptive chunking achieves remarkable results, which reveals the potential of adaptive chunking robust to prompts and sampling timesteps, further highlighting the superiority of our chunk-level optimization.
>
> We will do more exploration following this direction and incorperate it into the paper. Many thanks for your suggestions!

---

> ### Author Response · Authors · 2025-11-28
> **Part 2/2**
>
> # W3: Motivation Issues
> > Despite the insight the authors claimed in Figure 1 $\cdots$ its $t=1$ timestep is worse than that in $chunk_2$.
>
> Thank you for your careful review. We suspect there may be a slight misunderstanding regarding the group comparison in our method, and we would like to clarify this point.
>
> Your comment mentions comparing "$chunk_1$ has the greater final reward (advantage), its t=1 timestep is worse than that in $chunk_2$", which suggests a comparison between different chunks within the same trajectory. However, Chunk-GRPO (as well as standard GRPO) compares the chunks with same chunk index across different trajectories within a group, but not between different chunks in a same trajecotry. By comparing across trajectories, we increase the likelihood of the policy that produced the better outcome for that specific chunk interval and decrease the likelihood for the worse one. We do not compare $Chunk_1$ against $Chunk_2$ within a single image generation.
>
> Could you please clarify if we have understood your concern correctly? If you intended a different opinion, we would be happy to address it.
>
> Once again, thank you sincerely for your time, encouragement, and constructive feedback. Your suggestion regarding adaptive chunking was particularly valuable and has significantly strengthened our methodology. We promise that we will do more exploration following this direction and incorperate it into our paper. In terms of Weakness 3, we kindly ask further clarification if we understood wrong.
>
> # References
> [1] Wang, Y., Li, Z., Zang, Y., Zhou, Y., Bu, J., Wang, C., Lu, Q., Jin, C. and Wang, J., 2025. Pref-grpo: Pairwise preference reward-based grpo for stable text-to-image reinforcement learning. arXiv preprint arXiv:2508.20751.
>
> [2] Ghosh, Dhruba, Hannaneh Hajishirzi, and Ludwig Schmidt. "Geneval: An object-focused framework for evaluating text-to-image alignment." Advances in Neural Information Processing Systems 36 (2023): 52132-52152.
>
> [3] Radford, A., Kim, J.W., Hallacy, C., Ramesh, A., Goh, G., Agarwal, S., Sastry, G., Askell, A., Mishkin, P., Clark, J. and Krueger, G., 2021, July. Learning transferable visual models from natural language supervision. In International conference on machine learning (pp. 8748-8763). PmLR.

---

> ### Author Response · Authors · 2025-12-03
> **Update on the Adaptive Chunking**
>
> We have incorperated the adaptive chunking into the Appendix C.

---

### Official Review · Reviewer_ZLe3 · 2025-11-02

**Soundness:** 2
**Presentation:** 2
**Contribution:** 2
**Rating:** 4
**Confidence:** 3

**Summary:**

This paper proposes a method that involves fine-tuning the image generation process at the chunk level instead of at separate timesteps. Furthermore, it introduces a weighted sampling strategy derived from the proposed chunk split design.

**Strengths:**

1. The core idea of optimizing at the chunk level instead of at separate timesteps is an interesting and novel direction for image generation.
2. The proposed method is explained with clarity.

**Weaknesses:**

1. Proposition 1 and its corresponding proof raise concerns.
Specifically, Proposition 1 claims that a smaller chunk size leads to better performance. Since sampling through separate timesteps is equivalent to sampling with a chunk size set to $K=1$, the authors should further clarify the optimal limit for "how small is enough."

2. The proof of Proposition 1 also presents issues.
Eq. (35) states that $J_{\text{chunk}} = \frac{1}{T}J_{\text{GRPO}}$, suggesting the objective function for the proposed chunk split is simply a scaled version of the original GRPO objective.
While the optimal parameters $\theta_{\text{chunk}} = \arg\max_{\theta} J_{\text{chunk}}(\theta)$ and $\theta_{\text{GRPO}} = \arg\max_{\theta} J_{\text{GRPO}}(\theta)$ would be mathematically equal due to scaling, the proof's objective should focus on demonstrating how the policy parameters $\theta$ are affected by the chunking scheme, rather than comparing the squared errors in the form $\|\hat{J}(\theta) - J_{\text{GRPO}}(\theta)||^{2} \geq \|\hat{J}(\theta) - J_{\text{chunk}}(\theta)||^{2}$. The change in the optimal parameter $\theta$ should be explicitly shown.

Minors:
1. Typo errors were noted in some equations (e.g., in **Eq. (18)**, the notation should likely be $T_a \cup T_{ia} = \{1, 2, \cdots, T\}$).
2. The experimental results appear to show **only a marginal improvement** over the current state-of-the-art method, **Dance-GRPO**.

**Questions:**

1. Could the authors provide a more detailed justification for the selected chunk sizes of $[2, 3, 4, 7]$? (e.g provide the specific details of the $\ell_1(x, t)$ values across all timesteps to validate the chunk split design)
2. Could the authors elaborate on why the final attained policy parameters $\theta$ yield superior results? Is this improvement primarily attributable to enhanced stability in the reinforcement learning training process, or is there another underlying mechanism?

---

> ### Author Response · Authors · 2025-11-28
> **Part 1/2**
>
> Dear Reviewer ZLe3,
>
> We sincerely appreciate your thoughtful feedback and encouraging recognition of our novel idea and well-explained methodology. Your careful reading and insightful questions have been invaluable to improving our work. Below, we address each of your concerns in detail.
>
> # W1: The Definition of "small"
> > Proposition 1 and its corresponding proof $\cdots$ optimal limit for "how small is enough."
>
> Thank you for this thoughtful question. To clarify, the chunk size $2 \leq K \leq 4$ strictly satisfies the requirement derived in Equation (39). We identify $K=2$ as the optimal lower limit.
>
> Our conclusion that the chunk size should be "small" is based on the solution to Equation (39):
> $$m- \sqrt{m^2+3}+2 \leq T \leq m+ \sqrt{m^2+3}+2.$$
> It is true that setting the chunk size to $K=1$, where $T=m=1$ makes the first inequality into an equality. In this case, sampling through individual steps is equivalent to sampling with chunks. However, as long as $2 \leq T \leq 4$, the first and second inequality both always hold (and no equality) since we ssume $ m \geq 1$. Therefore, $K=2$ is the optimal limit.
>
> # W2 and Q2: Superiority of Chunking
> > The proof of Proposition 1 $\cdots$ The change in the optimal parameter $\theta$ should be explicitly shown?
>
> > Could the authors elaborate on why the final attained policy parameters $\theta$ or is there another underlying mechanism?
>
> We thank the reviewer for this insightful comment. While Equation (35) indicates that Chunk-GRPO's objective is a first-order approximation of the standard GRPO's objective, this does not imply the optimization dynamics are identical in practice. The key difference lies in their gradients, which makes Chunk-GRPO more stable.
>
> To begin with, the gradient of the GRPO's objective in Equation (5) can be derived as follows (clipping and KL are omitted for brevity):
> $$\begin{aligned}
>     \nabla_\theta J_{GRPO}(\theta) &= \nabla_\theta E_{c, x} \left[\frac{1}{G} \frac{1}{T} \sum_{i=1}^{G} \sum_{t=1}^{T} r_t^i \left(\theta \right)A^i \right] \\\\\\
>     &= \nabla_\theta E_{c, x} \left[\frac{1}{G} \sum_{i=1}^{G} A^i \frac{1}{T} \sum_{t=1}^{T} r_t^i \left(\theta \right)
>     \cdot \nabla_\theta \text{log} \left( r_t^i \left(\theta \right) \right) \right].
> \end{aligned}$$
> Given that
> $$r^i_t(\theta) = \frac{p_\theta (x^i_{t-1} | x^i_t, c)}{p_\text{old} (x^i_{t-1} | x^i_t, c)},$$
> this simplifies to:
> $$\begin{aligned}
>     \nabla_\theta J_{GRPO}(\theta) &= \nabla_\theta E_{c, x} \left[\frac{1}{G} \sum_{i=1}^{G} A^i \frac{1}{T} \sum_{t=1}^{T} \frac{p_\theta (x^i_{t-1} | x^i_t, c)}{p_\text{old} (x^i_{t-1} | x^i_t, c)}
>     \cdot \nabla_\theta \text{log} \left( p_\theta (x^i_{t-1} | x^i_t, c) \right) \right].
> \end{aligned}$$
> In comparision, the gradient of our Chunk-GRPO's objective in Equation (13) is derived below. For convenience, we denote the chunk-level importance ratio from Equation (14) as:
> $$s_j^i(\theta) = \left( \prod_{t \in ch_j} \frac{p_\theta \left( x_{t-1}^i | x_t^i, c \right)}{p_{\theta_{old}} \left( x_{t-1}^i | x_t^i, c \right)} \right) ^{\frac{1}{cs_j}},$$
> Then the gradient is:
> $$\begin{aligned}
>     \nabla_\theta J_{Chunk-GRPO}(\theta) &= \nabla_\theta E_{c, x} \left[\frac{1}{G} \frac{1}{K} \sum_{i=1}^{G} \sum_{j=1}^{K} s_j
>     ^i \left(\theta \right)A^i \right] \\\\\\
>     &= \nabla_\theta E_{c, x} \left[\frac{1}{G} \sum_{i=1}^{G} A^i \frac{1}{K} \sum_{j=1}^{K} s_j^i \left(\theta \right)
>     \cdot \nabla_\theta \text{log} \left( s_j^i \left(\theta \right) \right) \right] \\\\\\
>     &= \nabla_\theta E_{c, x} \left[\frac{1}{G} \sum_{i=1}^{G} A^i \frac{1}{K} \sum_{j=1}^{K} \left( \prod_{t \in ch_j} \frac{p_\theta \left( x_{t-1}^i | x_t^i, c \right)}{p_{\theta_{old}} \left( x_{t-1}^i | x_t^i, c \right)} \right) ^{\frac{1}{cs_j}}
>     \cdot \frac{1}{cs_j} \sum_{t \in ch_j}
>     \nabla_\theta \text{log} \left( p_\theta (x^i_{t-1} | x^i_t, c) \right) \right]
> \end{aligned}$$
> Therefore, the  fundamental distinction between ours and GRPO lies in how they weight the gradients of the log likelihoods of tokens. In standard GRPO, tokens are weighted individually according to their respective importance weight $\frac{p_\theta (x^i_{t-1} | x^i_t, c)}{p_\text{old} (x^i_{t-1} | x^i_t, c)}$. However, these unequal weights, which can vary among $(0,1 + \epsilon]$ for $A \geq 0$ or $[1 - \epsilon, \infty)$ for $A \leq 0$, are not negligible, and their impact can accumulate and lead to unstable consequences. In contrast, our approach applies a unified weight $\left( \prod_{t \in ch_j} \frac{p_\theta \left( x_{t-1}^i | x_t^i, c \right)}{p_{\theta_{old}} \left( x_{t-1}^i | x_t^i, c \right)} \right) ^{\frac{1}{cs_j}}$ to all tokens within a chunk, effectively smoothing these fluctuations and eliminating this instability.

---

> ### Author Response · Authors · 2025-11-28
> **Part 2/2**
>
> # M(inors)1: Typos
> > Typo errors $\cdots$
>
> Thank you for catching these. We have thoroughly reviewed the manuscript and corrected all typos to ensure clarity. We apologize for any disruption to your reading experience.
>
> # M2: Marginal Improvement
> > The experimental results appear to show only a marginal improvement $\cdots$
>
> Thank you for pointing this out. In our original report, Chunk-GRPO demonstrates significant improvements in preference alignment, while moderate gains on the text-to-image (T2I) benchmark, WISE.
>
> To further validate Chunk-GRPO's effectiveness, we conducted experiments on the standard T2I benchmark GenEval[1], using CLIP[2] as the reward model.
>
> **Table 5.** Results on GenEval.
> | Model | Single Obj. | Two Obj. | Counting | Colors | Position | Color Attri. | Overall |
> | ----- | ----------- | -------- | -------- | ------ | -------- | ------------ | ------- |
> | Flux | 0.99 | 0.83 | 0.71 | 0.75 | **0.24** | 0.44 | 0.66 |
> | Dance-GRPO | **1.00** | **0.86** | 0.71 | 0.78 | 0.22 | 0.46 | 0.67 |
> | Chunk-GRPO w/o ws | 0.99 | 0.85 | **0.75** | **0.81** | 0.21 | **0.51** | **0.69** |
>
> Results in Table 5 demonstrate that Chunk-GRPO outperforms the standard step-level GRPO (Dance-GRPO) significantly. Specifically, Chunk-GRPO achieves a performance gain of 0.03, which is $3$ times larger than the gain achieved by Dance-GRPO.
>
> We attribute the moderate improvement on WISE to a misalignment between the evaluation task and the reward model. Specifically, WISE focuses on reasoning capabilities in T2I generation, whereas HPSv3 is a reward model optimized primarily for aesthetic alignment. As a proof, the results in Table 1 demonstrate the significant outperformance of our method in aesthetic preference alignment.
>
> We have incorperated these into the paper.
>
> # Q1: Chunk Selection
> > Could the authors provide a more detailed justification for the selected chunk sizes $\cdots$
>
> Thank you for raising this point. As illustrated in Figure 8, the relative $L1$ distance of Flux is:
> $$[0.748, 0.771, 0.534, 0.445, 0.403, 0.376, 0.359, 0.347, 0.340, 0.337, 0.334, 0.333, 0.331, 0.326, 0.315, 0.299];$$
> After training with the baseline Dance-GRPO, these shift to:
> $$[0.748, 0.770, 0.534, 0.449, 0.402, 0.376, 0.359, 0.348, 0.342, 0.343, 0.343, 0.348, 0.353, 0.356, 0.355, 0.341].$$
> It is cleary observed that the trends in the last 7 timesteps deviate significantly after training with the baseline, therefore we grouped these into a chunk. Additionly, the first 2 timesteps naturally form a distinct chunk. For the remaining middle timesteps, we analyzed their first-order and second-order derivatives. Using a second-order derivative threshold of $0.01$, we split the remaining steps into 2 chunks.
>
> Once again, thank you sincerely for your time and constructive feedback throughout the review process. Your insightful suggestions have been invaluable in strengthening our manuscript.
>
> # References
> [1] Ghosh, Dhruba, Hannaneh Hajishirzi, and Ludwig Schmidt. "Geneval: An object-focused framework for evaluating text-to-image alignment." Advances in Neural Information Processing Systems 36 (2023): 52132-52152.
>
> [2] Radford, A., Kim, J.W., Hallacy, C., Ramesh, A., Goh, G., Agarwal, S., Sastry, G., Askell, A., Mishkin, P., Clark, J. and Krueger, G., 2021, July. Learning transferable visual models from natural language supervision. In International conference on machine learning (pp. 8748-8763). PmLR.

---

### Author Response · Authors · 2025-12-03
**Summary of the Paper, Rebuttal, and Manuscript Revisions (Part 1/2)**

Dear Area Chair and Reviewers:

We would like to sincerely thank you once again for your time, effort, and valuable insights on our work. Your constructive comments and thoughtful suggestions have been extremely helpful. Below is a concise summary of our paper, rebuttal and the key manuscript revisions points for your ease of reference.

# The Paper

This paper proposed Chunk-GRPO, the **first chunk-level GRPO-based approach for flow-matching-based text-to-image (T2I) generation**. By grouping consecutive timesteps into coherent "chunk"s that capture the intrinsic temporal dynamics of flow matching and optimizing policies at the chunk level, we **novelly shift the GRPO optimization from the original step level to chunk level**, which **mitigates two key limitations** in GRPO: (1) inaccurate advantage assignments, and (2) the overlook of temporal dynamics. Compared to the original GRPO, our method effectively smooths the fluctuations and eliminating instability during training. Chunk-GRPO also incorporates an optional  weighted sampling strategy which emphasizes high-noise chunks, achieving improvements on standard T2I benchmarks like GenEval as well as preference alignments like HPS.

# Reviewer Highlights from the Original Review

Our paper has been acknowledged for its **insightful motivation, strong novelty, principled methodology, clean theoretical justification, comprehensive empirical evaluation, and well-organized presentation**. Notable highlights include:

- Our motivation of addressing the inaccurate advantage attributions shown in Figure 1 is very **interesting** and **insightful** (Reviewer VUdY), and it points to the **core limitations** of GRPO (Reviewer xpH9).

- Our idea of grouping consecutive timesteps into chunks guided by prompt-invariant relative $L1$ latent dynamics is **novel**, **interesting**, and **worthy of further exploration (All Reviewers)**.

- Our method of using the chunk-level importance ratio with the chunk segmentation guided by temporal dynamics is **principled and general without arbitrary chunking or hyperparameter configurations** (Reviewer VUdY, K7kg). It is also **explained with clarity** (Reviewer ZLe3) and **straightforward with easy implementation**(Reviewer xpH9). Moreover, our findings on temporal dynamics are **beneficial to the community** (Reviewer VUdY).

- Our mathematical analysis showing why chunk-level optimization yields better and smoother training results is quite **clear** (Reviewer xpH9).

- Our empirical evaluation demonstrating consistent gains in preference alignment and competitive T2I benchmark is **comprehensive** (Reviewer xpH9) with **extensive ablation studies** (Reviewer VUdY).

---

> ### Author Response · Authors · 2025-12-03
> **Summary of the Paper, Rebuttal, and Manuscript Revisions (Part 2/2)**
>
> # Addressing the Raised Concerns
>
> We have **carefully reviewed and addressed all concerns**, including **common concerns**:
>
> - Marginal Improvement (Minors1 of Reviewer ZLe3, W1 of Reviewer VUdY, W3 of Reviewer k7kg, W3 of Reviewer xpH9): We have conducted experiments on the standard T2I benchmark **GenEval**, demonstrating that Chunk-GRPO outperforms the standard step-level GRPO (Dance-GRPO) **significantly with a 3 times larger gains**. These experiment results further validate Chunk-GRPO's effectiveness. We also gave **analysis of the moderate improvement on WISE** (because of the misalignment between the evaluation task and the reward model).
>
> - Heuristic Chunking (W2 of Reviewer VUdY, Q1 of Reviewer xpH9): While our initial implementation used fixed chunks, we have developed an **adaptive chunking strategy robust to prompts and sampling timesteps**.  The adaptive chunking **achieves remarkable results**, which reveals the potential of it, further highlighting the superiority of our chunk-level optimization.
>
> as well as **indivisual concerns**:
>
> - More Theoretical Justification (W1, W2 and Q2 of Reviewer ZLe3): In addition to our initial theoretical analysis, we have further given an **underlying mechanism from the perspective of gradients weights** to demonstrate why Chunk-GRPO offers a more stable training process. We also gave a more **detailed definition of "small chunk"** of Proposition 1.
>
> - More Chunk Selection Details (Q1 of Reviewer ZLe3): We have demonstrate a **detailed justification for our initial preactice** of the selected chunk sizes $[2, 3, 4, 7]$.
>
> - More Baselines (W1 of Reviewer VUdY): We have conducted an **additional comparision with Pref-GRPO** as the baseline, confirming the effectiveness of our chunk-level optimization.
>
> - Added User Studies (W2 of Reviewer xpH9): We have conducted a study to assess human preferences. Our Chunk-GRPO is **preferred by human reviewers $72.5$% of the time**, which further demonstrates the significant superiority of our approach.
>
> - Added Significance Analysis (W1 of Reviewer xpH9): We have **reported the standard deviations** in our experiments.
>
> - Typos (Minors2 of Reviewer ZLe3): We have thoroughly reviewed the manuscript and corrected all typos to ensure clarity.
>
> Moreover, we suspect that there may be some **slight misunderstandings** regarding the group comparison and advantages comparision in our method, and we have **clarified these points** (W3 of Reviewer VUdY, W1 and W2 of Reviewer k7kg).
>
> # Revisions of the Manuscript
>
> We have incorporated some revisions to our manuscript, including:
>
> - Experiments on Geneval: Table 5
>
> - Pref-GRPO as a Baseline: Table 7 in the Appendix
>
> - Adaptive Chunking: Appendix C.5 and Table 8
>
> - User Studies: Table 9 in the Appendix
>
> - Additional Theoretical Justification: the second part of Appendix A from Line 827.
>
> - We added Figure 1 and modified Figure 2, 3 and 4 (previous Figure 1, 2, and 3) for clearer presentation.
>
> Finally, we are truly grateful for the reviewers' constructive suggestions, which have significantly improved our work. We sincerely thank you once again for your time, careful review, and valuable feedback. Moreover, **in terms of the misunderstanding suspection**, due to this year's unexpected early stopping of discussion, we sincerely encourage Reviewers VUdY and Reviewers k7kg for **further discussions after the paper decision**. We feel really sorry for that.

---

### Meta-Review · Area_Chair_p7tJ · 2026-01-05

**Summary:**

This paper proposed chunk-level GRPO-based approach for T2I generation. Authors also introduced an optional weighted sampling strategy to further enhance performance. Experimental results show effectiveness of the proposed method.

This paper got three 4 ratings and one 6 rating.

The strength of this paper given by reviewers are:
1. idea is interesting and novel. (Reviewer ZLe3, VUdY, k7kg, xpH9)
2. proposed method is explained with clarity. (Reviewer ZLe3)
3. a principled, dynamics-aware segmentation. (Reviewer VUdY, k7kg)
4. paper is well-organized and easy to follow. (Reviewer xpH9)
5. a clean mathematical analysis. (Reviewer xpH9)
6. extensive experiments. (Reviewer xpH9)

The weakness of this paper given by reviewers are:
1. Proposition 1 and its corresponding proof raise concerns. (Reviewer ZLe3)
2. The proof of Proposition 1 also presents issues. (Reviewer ZLe3)
3. Typo errors were noted in some equations. (Reviewer ZLe3)
4. The experimental results appear to show only a marginal improvement over the current state-of-the-art method, Dance-GRPO. (Reviewer ZLe3, VUdY, k7kg, xpH9)
5. does not compare against other GRPO variants like Flow-GRPO, Pref-GRPO. (Reviewer VUdY)
6. chunking implementation is heuristic. (Reviewer VUdY)
7. chunk-based design did not show an optimal approach to solving such issues. (Reviewer VUdY)
8. More intuitive results. (Reviewer k7kg)
9. Motivation concern. (Reviewer k7kg)
10. missing significance analysis. (Reviewer xpH9)
11. user study is missing. (Reviewer xpH9)

Questions:
1. provide a more detailed justification for the selected chunk sizes. (Reviewer ZLe3)
2. elaborate on why the final attained policy parameters yield superior results? (Reviewer ZLe3)
3. adaptive chunking. (Reviewer xpH9)

AC carefully read authors' paper, reviewers' comments and authors' rebuttal. AC found authors didn't fully addressed reviewers' concerns and have to reject this paper. The details are in the session below.

**Reviewer Concerns:**

weakness 1. it is still not clear why K=2 is better than K=1.

weakness 2. authors added more explanation.

weakness 3. authors fixed it.

weakness 4. the results is still a bit mixed.

weakness 5. Flow-GRPO is not discussed. also why Table 5 didn't include results from Perf-GRPO?

weakness 6. authors added more results.

weakness 7. seems there still some misunderstanding. unfortunately, there is no further discussion.

weakness 8. authors added more explanation.

weakness 9. authors added more explanation.

weakness 10. authors added significance analysis, through 3 random seeds might not be enough.

weakness 11. authors added user study.


question 1. authors provided details.

question 2. not answered.

question 3. authors added more results.

**Reviewer Scores:**

Reviewer ZLe3 might keep their rating 4.

Reviewer VUdY might keep their rating 4.

Reviewer k7kg might keep their rating 4.

Reviewer xpH9 might keep their rating 6.

---

### Decision · Program_Chairs · 2026-01-26

Reject